# Efficient Diffusion Policies for Offline Reinforcement Learning

**Bingyi Kang**[*] **Xiao Ma**[*] **Chao Du** **Tianyu Pang** **Shuicheng Yan**
Sea AI Lab
{bingykang,yusufma555,duchao0726}@gmail.com {tianyupang,yansc}@sea.com

## Abstract

Offline reinforcement learning (RL) aims to learn optimal policies from offline datasets, where the parameterization of policies is crucial but often overlooked. Recently, Diffsuion-QL [37] significantly boosts the performance of offline RL by representing a policy with a diffusion model, whose success relies on a parametrized Markov Chain with hundreds of steps for sampling. However, Diffusion-QL suffers from two critical limitations. 1) It is computationally inefficient to forward and backward through the whole Markov chain during training. 2) It is incompatible with maximum likelihood-based RL algorithms (*e.g.*, policy gradient methods) as the likelihood of diffusion models is intractable. Therefore, we propose *efficient diffusion policy* (EDP) to overcome these two challenges. EDP approximately constructs actions from corrupted ones at training to avoid running the sampling chain. We conduct extensive experiments on the D4RL benchmark. The results show that EDP can reduce the diffusion policy training time **from 5 days to 5 hours** on gym-locomotion tasks. Moreover, we show that EDP is compatible with various offline RL algorithms (TD3, CRR, and IQL) and achieves new state-of-the-art on D4RL by large margins over previous methods. Our code is available at https://github.com/sail-sg/edp.

## 1 Introduction

Offline reinforcement learning (RL) is much desired in real-world applications as it can extract knowledge from previous experiences, thus avoiding costly or risky online interactions. Extending online RL algorithms to the offline domain faces the distributional shift [9] problem. Existing methods mainly focus on addressing this issue by constraining a policy to stay close to the data-collecting policy [6, 39], making conservative updates for Q-networks [17, 15, 40], or combining these two strategies [21, 12]. However, Offline RL can also be viewed as a state-conditional generative modeling problem of actions, where the parameterization of the policy network is important but largely overlooked. Most offline RL works follow the convention of parameterizing the policy as a diagonal Gaussian distribution with the learned mean and variance. This scheme might become inferior when the data distribution is complex, especially when offline

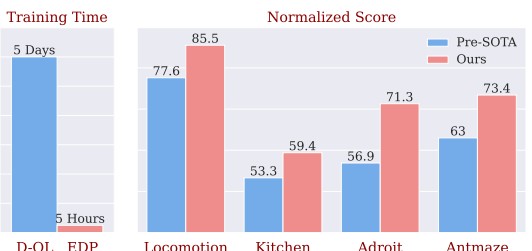

Figure 1: Efficiency and Generality. D-QL is Diffusion-QL. *Left*: The training time on the locomotion tasks in D4RL. *Right*: the performance of EDP and previous SOTA on each domain in D4RL. EDP is trained with TD3 on locomotion and IQL on the other three domains. (Best viewed in color.)

---

[*]equal contribution

37th Conference on Neural Information Processing Systems (NeurIPS 2023).

data are collected from various sources and present strong multi-modalities [32]. Therefore, more expressive models for the policy are strongly desired.

Recently, Diffusion-QL [37] made a successful attempt by replacing the diagonal Gaussian policy with a diffusion model, significantly boosting the performance of the TD3+BC [6] algorithm. Diffusion models [34, 10] have achieved the new state-of-the-art (SOTA) in image generation tasks [26, 5], demonstrating a superior ability to capture complex data distributions.

Despite the impressive improvement that Diffusion-QL has achieved, it has two critical drawbacks preventing it from practical applications. *First*, training a diffusion policy with offline RL is computationally inefficient. Consider a parameterized diffusion policy $\pi_\theta(\boldsymbol{a}|\boldsymbol{s})$, Diffusion-QL optimizes it by maximizing the Q value $Q(\boldsymbol{s}, \boldsymbol{a}_\theta)$ of a state $\boldsymbol{s}$ given a policy-generated action $\boldsymbol{a}_\theta \sim \pi_\theta(\boldsymbol{a} \mid \boldsymbol{s})$. However, sampling from a diffusion model relies on a long parameterized Markov chain (*e.g.*, 1,000 steps), whose forward inference and gradient backpropagation are unaffordably expensive. *Second*, diffusion policy is not a generic policy class as it is restricted to TD3-style algorithms. As computing the sample likelihood $\pi_\theta(a \mid s)$ is intractable in diffusion models [35], diffusion policy is incompatible with a large family of policy gradient algorithms (*e.g.*, V-Trace [22], AWR [28], IQL [15]), which require a tractable and differentiable log-likelihood $\log \pi_\theta(\boldsymbol{a}|\boldsymbol{s})$ for policy improvement.

In this work, we propose *efficient diffusion policy* (EDP) to address the above two limitations of diffusion policies. Specifically, we base EDP on the denoising diffusion probabilistic model (DDPM) [10], which learns a noise-prediction network to predict the noise used to corrupt an example. In the forward diffusion process, a corrupted sample follows a predefined Gaussian distribution when the clean example and timestep are given. In turn, given a corrupted sample and predicted noise, we can approximate its clean version by leveraging the reparametrization trick. Based on this observation, to avoid the tedious sampling process, we propose *action approximation* to build an action from a corrupted one, which can be easily constructed from the dataset. In this way, each training step only needs to pass through the noise-prediction network once, thus substantially reducing the training time. As experimented, by simply adding action approximation, we obtain **2x** speed-up without performance loss. Moreover, we apply DPM-Solver [20], a faster ODE-based sampler, to further accelerate both the training and sampling process. Finally, to support likelihood-based RL algorithms, we leverage the evidence lower bound for the likelihood developed in DDPM and approximate the policy likelihood from a constructed Gaussian distribution with variance fixed and mean obtained from action approximation.

We evaluate the efficiency and generality of our method on the popular D4RL benchmarking, as shown in Fig. 1. We first benchmark the efficiency of EDP on gym-locomotion tasks. By replacing the diffusion policy in Diffusion-QL with our EDP, the training time of Diffusion-QL is reduced substantially **from five days to five hours** (compared to their official code). Meanwhile, we observe slight to clear performance improvements on different tasks as the improved efficiency enables training DDPM with more timesteps than before. Moreover, we plug EDP into three different offline algorithms (including TD3+BC, CRR, and IQL), and the results justify its superiority over standard diagonal Gaussian policies. As a result, EDP set up new state-of-the-art on all four domains in D4RL.

## 2 Related Work

**Offline RL** Distributional shift between the learned and behavior policies is offline RL's biggest challenge. Existing research mitigates this problem by making modifications to policy evaluation [24, 17, 15, 21] or policy improvement [9, 39, 6, 33, 38, 42, 41]. For example, conservative Q-learning (CQL) [17] penalizes out-of-distribution actions for having higher Q-values, proving that this is equivalent to optimizing a lower bound of Q-values. Onestep RL [1] conducts policy evaluation on in-distribution data to avoid querying unseen actions. IQL [15] introduces expectile regression [14] to approximate dynamic programming with the Bellman optimality function. TD3+BC explicitly constrains the learned policy by adding a behavior cloning loss to mimic the behavior policy. Instead, CRR and AWR impose an implicit policy regularization by performing policy gradient-style policy updates. Despite their effectiveness, they ignore that the capacity of a policy representation plays a vital role in fitting the data distribution. This paper instead focuses on an orthogonal aspect (*i.e.*, policy parameterization) that all the above methods can benefit. Another line of work tries to cast offline RL as a sequence-to-sequence model [3, 11], which is beyond the scope of this work.

**Policy Parametrization** Different RL algorithms may pose different requirements for parameterizing a policy distribution. There are mainly two categories of requirements: 1) The sampling process

is differentiable, such as the deterministic policy in DDPG [18] and TD3 [7]. 2) The log-likelihood of samples is tractable. For example, policy gradient methods [30, 31, 38, 28] optimize a policy based on maximum likelihood estimation (MLE). Therefore, most works represent policy with a diagonal Gaussian distribution with mean and variance parameterized with a multi-layer perceptron (MLP). On the other hand, BCQ [9] and BEAR [16] choose to model policy with a conditional variational autoencoder (CVAE). Recently, Diffusion-QL [37] introduced diffusion models into offline RL and demonstrated that diffusion models are superior at modeling complex action distributions than CVAE and diagonal Gaussian. However, it takes tens to hundreds more time to train a diffusion policy than a diagonal Gaussian one. Moreover, diffusion policy only satisfies the first requirement, which means many other offline RL algorithms can not use it, including the current SOTA IQL.

Our method is motivated to solve the about two limitations. We first propose a more efficient way to train diffusion policies, which reduces training time to the level of a Gaussian policy. Then, we generalize the diffusion policy to be compatible with MLE-based RL methods.

## 3 Preliminaries

### 3.1 Offline Reinforcement Learning

A decision-making problem in reinforcement learning is usually represented by a Markov Decision Process (MDP): $\mathcal{M} = \{\mathcal{S}, \mathcal{A}, P, R, \gamma\}$. $\mathcal{S}$ and $\mathcal{A}$ are the state and action spaces respectively, $P(s'|s, a)$ measures the transition probability from state $s$ to state $s'$ after taking action $a$ while $R(s, a, s')$ gives the reward for the corresponding transition, $\gamma \in [0, 1)$ is the discount factor. A policy $\pi(a|s)$[2] describes how an agent interacts with the environment. The optimal policy $\pi^*(a|s)$ is the one achieves maximal cumulative discounted returns: $\pi^* = \arg\max \mathbb{E}_\pi \left[ \sum_{t=0}^{\infty} \gamma^t r(s_t, a_t) \right]$. Reinforcement learning algorithms frequently rely on the definition of value function $V(s) = \mathbb{E}_\pi \left[ \sum_{t=0}^{\infty} \gamma^t r(s_t, a_t) | s_0 = s \right]$, and action value (Q) function $Q(s, a) = \mathbb{E}_\pi \left[ \sum_{t=0}^{\infty} \gamma^t r(s_t, a_t) | s_0 = s, a_0 = a \right]$, which represents the expected cumulative discounted return of a policy $\pi$ given the initial state $s$ or state-action pair $(s, a)$.

In the offline RL setting, instead of learning from interactions with the environment, agents focus on learning an optimal policy from a previously collected dataset of transitions: $\mathcal{D} = \{(s_t, a_t, s_{t+1}, r_t)\}$. Offline RL algorithms for continuous control are usually based on an actor-critic framework that alternates between policy evaluation and policy improvement. During policy evaluation, a parameterized Q network $Q_\phi(s, a)$ is optimized based on approximate dynamic programming to minimize the following temporal difference (TD) error $L_{\text{TD}}(\phi)$:

$\mathbb{E}_{(s,a,s')\sim\mathcal{D}} \left[ \left( r(s, a) + \gamma \max_{a'} Q_{\hat{\phi}}(s', a') - Q_\phi(s, a) \right)^2 \right]$, where $Q_{\hat{\phi}}(s, a)$ denotes a target network. Then at the policy improvement step, knowledge in the Q network is distilled into the policy network in various ways. Offline RL methods address the distributional shift [9] problem induced by the offline dataset $\mathcal{D}$ by either modifying the policy evaluation step to regularize Q learning or constraining the policy improvement directly. In the following, we will show that our diffusion policy design is compatible with any offline algorithms and can speed up policy evaluation and improvement.

### 3.2 Diffusion Models

Consider a real data distribution $q(x)$ and a sample $x^0 \sim q(x)$ drawn from it. The (forward) diffusion process fixed to a Markov chain gradually adds Gaussian noise to the sample in $K$ steps, producing a sequence of noisy samples $x^1, \ldots x^K$. Note that we use superscript $k$ to denote diffusion timestep to avoid conflicting with the RL timestep. The noise is controlled by a variance schedule $\beta^1, \ldots, \beta^K$:

$$q(x^k|x^{k-1}) = \mathcal{N}(x^k; \sqrt{1-\beta^k}x^{k-1}, \beta^k I), \quad q(x^{1:K}|x^0) = \prod_{k=1}^{K} q(x^k|x^{k-1}). \quad (1)$$

When $K \to \infty$, $x^K$ distributes as an isotropic Gaussian distribution. Diffusion models learn a conditional distribution $p_\theta(x^{t-1}|x^t)$ and generate new samples by reversing the above process:

$$p_\theta(x^{0:K}) = p(x^K)\prod_{k=1}^{K} p_\theta(x^{k-1}|x^k), \quad p_\theta(x^{k-1}|x^k) = \mathcal{N}(x^{k-1}; \mu_\theta(x^k, k), \Sigma_\theta(x^k, k)), \quad (2)$$

---

[2]A policy could either be deterministic or stochastic, we use its stochastic form without loss of generality.

where $p(\boldsymbol{x}^K) = \mathcal{N}(\boldsymbol{0}, \boldsymbol{I})$ under the condition that $\prod_{k=1}^{K}(1 - \beta^k) \approx 0$. The training is performed by maximizing the evidence lower bound (ELBO):$\mathbb{E}_{\boldsymbol{x}_0}[\log p_\theta(\boldsymbol{x}^0)] \geq \mathbb{E}_q\left[\log \frac{p_\theta(\boldsymbol{x}^{0:K})}{q(\boldsymbol{x}^{1:K}|\boldsymbol{x}^0)}\right]$.

# 4 Efficient Diffusion Policy

In this section, we detail the design of our efficient diffusion policy (EDP). First, we formulate an RL policy with a diffusion model. Second, we present a novel algorithm that can train a diffusion policy efficiently, termed Reinforcement-Guided Diffusion Policy Learning (RGDPL). Then, we generalize the diffusion policy to work with arbitrary offline RL algorithms and compare our EDP with Diffusion-QL to highlight its superiority in efficiency and generality. Finally, we discuss several methods to sample from the diffusion policy during evaluation.

## 4.1 Diffusion Policy

Following [37], we use the reverse process of a conditional diffusion model as a parametric policy:

$$\pi_\theta(\boldsymbol{a}|\boldsymbol{s}) = p_\theta(\boldsymbol{a}^{0:K}|\boldsymbol{s}) = p(\boldsymbol{a}^K)\prod_{k=1}^{K} p_\theta(\boldsymbol{a}^{k-1}|\boldsymbol{a}^k, \boldsymbol{s}), \tag{3}$$

where $\boldsymbol{a}^K \sim \mathcal{N}(\boldsymbol{0}, \boldsymbol{I})$. We choose to parameterize $\pi_\theta$ based on Denoising Diffusion Probabilistic Models (DDPM) [10], which sets $\boldsymbol{\Sigma}_\theta(\boldsymbol{a}^k, k; \boldsymbol{s}) = \beta^k \boldsymbol{I}$ to fixed time-dependent constants, and constructs the mean $\boldsymbol{\mu}_\theta$ from a noise prediction model as: $\boldsymbol{\mu}_\theta(\boldsymbol{a}^k, k; \boldsymbol{s}) = \frac{1}{\sqrt{\alpha^k}}\left(\boldsymbol{a}^k - \frac{\beta^k}{\sqrt{1-\bar{\alpha}^k}}\boldsymbol{\epsilon}_\theta(\boldsymbol{a}^k, k; \boldsymbol{s})\right)$, where $\alpha^k = 1 - \beta^k$, $\bar{\alpha}^k = \prod_{s=1}^{k}$, and $\boldsymbol{\epsilon}_\theta$ is a parametric model.

To obtain an action from DDPM, we need to draw samples from $K$ different Gaussian distributions sequentially, as illustrated in Eqn. (2)-(3). The sampling process can be reformulated as

$$\boldsymbol{a}^{k-1} = \frac{1}{\sqrt{\alpha^k}}\left(\boldsymbol{a}^k - \frac{\beta^k}{\sqrt{1-\bar{\alpha}^k}}\boldsymbol{\epsilon}_\theta(\boldsymbol{a}^k, k; \boldsymbol{s})\right) + \sqrt{\beta^k}\boldsymbol{\epsilon}, \tag{4}$$

with the reparametrization trick, where $\boldsymbol{\epsilon} \sim \mathcal{N}(\boldsymbol{0}, \boldsymbol{I})$, $k$ is the reverse timestep from $K$ to 0.

Similar to DDPM, plugging in the conditional Gaussian distributions, the ELBO in Sec. 3.2 can be simplified to the following training objective $L_{\text{diff}}(\theta)$:

$$\mathbb{E}_{k,\boldsymbol{\epsilon},(\boldsymbol{a}^0,\boldsymbol{s})}\left[\left\|\boldsymbol{\epsilon} - \boldsymbol{\epsilon}_\theta\left(\sqrt{\bar{\alpha}^k}\boldsymbol{a}^0 + \sqrt{1-\bar{\alpha}^k}\boldsymbol{\epsilon}, k; \boldsymbol{s}\right)\right\|^2\right], \tag{5}$$

where $k$ follows a uniform distribution over the discrete set $\{1, \ldots, K\}$. It means the expectation is taken over all diffusion steps from clean action to pure noise. Moreover, $\boldsymbol{\epsilon} \in \mathcal{N}(\boldsymbol{0}, \boldsymbol{I})$, and $(\boldsymbol{a}^0, \boldsymbol{s}) \in \mathcal{D}$ are state-action pairs drawn from the offline dataset. Given a dataset, we can easily and efficiently train a diffusion policy in a behavior-cloning manner as we only need to forward and backward through the network once each iteration. As shown in Diffusion-QL [37], diffusion policies can greatly boost the performance when trained with TD3-based Q learning. However, it still faces two main drawbacks that limit its real-world application: 1) It is inefficient in sampling and training; 2) It is not generalizable to other strong offline reinforcement learning algorithms.

## 4.2 Reinforcement-Guided Diffusion Policy Learning

To understand how a parametric policy $\pi_\theta$ is trained with offline RL algorithms, we start with a typical Q-learning actor-critic framework for continuous control, which iterates between policy evaluation and policy improvement. Policy evaluation learns a Q network by minimizing the TD error $L_{\text{TD}}(\phi)$:

$$\mathbb{E}_{(\boldsymbol{s},\boldsymbol{a},\boldsymbol{s}')\sim\mathcal{D}}\left[\left(r(\boldsymbol{s},\boldsymbol{a}) + \gamma Q_{\hat{\phi}}(\boldsymbol{s}',\boldsymbol{a}') - Q_\phi(\boldsymbol{s},\boldsymbol{a})\right)^2\right], \tag{6}$$

where the next action $\boldsymbol{a}' \sim \pi_\theta(\cdot|\boldsymbol{s}')$. The policy is optimized to maximize the expected Q values :

$$\max_\theta \mathbb{E}_{\boldsymbol{s}\sim\mathcal{D},\boldsymbol{a}\sim\pi_\theta(\boldsymbol{a}|\boldsymbol{s})}\left[Q_\phi(\boldsymbol{s},\boldsymbol{a})\right]. \tag{7}$$

It is straightforward to optimize this objective when a Gaussian policy is used, but things get much more difficult when a diffusion policy is considered due to its complicated sampling process. Instead, we propose to view the offline RL problem from the perspective of generative modeling, where a diffusion policy can be easily learned in a supervised manner from a given dataset. However, unlike in computer vision, where the training data are usually perfect, offline RL datasets often contain suboptimal state-action pairs. Suppose we have a well-trained Q network $Q_\phi$, the question becomes how we can efficiently use $Q_\phi$ to guide diffusion policy training procedure. We now show that this can be achieved without sampling actions from diffusion policies.

Let's revisit the forward diffusion process in Eqn. 1. A notable property of it is that the distribution of noisy action $a^k$ at any step $k$ can be written in closed form: $q(a^k|a^0) = \mathcal{N}(a^k; \sqrt{\bar{\alpha}^k}a^0, (1 - \bar{\alpha}^k)I)$. Using the reparametrization trick, we are able to connect $a^k$, $a^0$ and $\epsilon$ by:

$$a^k = \sqrt{\bar{\alpha}^k}a^0 + \sqrt{1 - \bar{\alpha}^k}\epsilon, \quad \epsilon \sim \mathcal{N}(0, I). \tag{8}$$

Recall that our diffusion policy is parameterized to predict $\epsilon$ with $\epsilon_\theta(a^k, k; s)$. By relacing $\epsilon$ with $\epsilon_\theta(a^k, k; s)$ and rearranging Eqn. (8), we obtain the approximiated action:

$$\hat{a}^0 = \frac{1}{\sqrt{\bar{\alpha}^k}}a^k - \frac{\sqrt{1 - \bar{\alpha}^k}}{\sqrt{\bar{\alpha}^k}}\epsilon_\theta(a^k, k; s). \tag{9}$$

In this way, instead of running the reverse diffusion process to sample an action $a^0$, we can cheaply construct $\hat{a}^0$ from a state-action pair $(s, a)$ in the dataset by first corrupting the action $a$ to $a^k$ then performing one-step denoising to it. We will refer to this technique as ***action approximation*** in the following. Accordingly, the policy improvement for diffusion policies is modified as follows:

$$L_\pi(\theta) = -\mathbb{E}_{s\sim\mathcal{D},\hat{a}^0}\left[Q_\phi(s, \hat{a}^0)\right]. \tag{10}$$

To improve the efficiency of policy evaluation, we propose to replace the DDPM sampling in Eqn. (4) with DPM-Solver [20], which is an ODE-based sampler. The algorithm is defered to the appendix.

## 4.3 Generalization to Various RL algorithms

There are mainly two types of approaches to realize the objective in Eqn. 7 for policy improvement.

*Direct policy optimization*: It maximizes Q values and directly backpropagate the gradients from Q network to policy network, *i.e.*, $\nabla_\theta L_\pi(\theta) = -\frac{\partial Q_\phi(s,a)}{\partial a}\frac{\partial a}{\partial \theta}$. This is only applicable to cases where $\frac{\partial a}{\partial \theta}$ is tractable, *e.g.*, when a deterministic policy $a = \pi_\theta(s)$ is used or when the sampling process can be reparameterized. Sample algorithms belonging to this category include TD3 [7], TD3+BC [6], and CQL [17]. One can easily verify that both the expensive DDPM sampling in Eqn. (4) and our efficient approximation in Eqn. (9) can be used for direct policy optimization.

*Likelihood-based policy optimization*: It tries to distill the knowledge from the Q network into the policy network indirectly by performing weighted regression or weighted maximum likelihood:

$$\max_\theta \quad \mathbb{E}_{(s,a)\sim\mathcal{D}}\left[f(Q_\phi(s, a))\log\pi_\theta(a|s)\right], \tag{11}$$

where $f(Q_\phi(s, a))$ is a monotonically increasing function that assigns a weight to each state-action pair in the dataset. This objective requires the log-likelihood of the policy to be tractable and differentiable. AWR [28], CRR [38], and IQL [15] fall into this category but each has a unique design in terms of the weighting function $f$. Since the likelihood of samples in Diffusion models is intractable, we propose the following two variants for realizing Eqn. 11.

First, instead of computing the likelihood, we turn to a lower bound for $\log\pi_\theta(a|s)$ introduced in DDPM [10]. By discarding the constant term that does not depend on $\theta$, we can have the objective:

$$\mathbb{E}_{k,\epsilon,(a,s)}\left[\frac{\beta^k \cdot f(Q_\phi(s, a))}{2\alpha^k(1 - \bar{\alpha}^{k-1})}\left\|\epsilon - \epsilon_\theta\left(a^k, k; s\right)\right\|^2\right]. \tag{12}$$

Second, instead of directlying optimizing $\log\pi_\theta(a|s)$, we propose to replace it with an approximated policy $\hat{\pi}_\theta(a|s) \triangleq \mathcal{N}(\hat{a}^0, I)$, where $\hat{a}^0$ is from Eqn. (9). Then, we get the following objective:

$$\mathbb{E}_{k,\epsilon,(a,s)}\left[f(Q_\phi(s, a))\left\|a - \hat{a}^0\right\|^2\right]. \tag{13}$$

Empirically, we find these two choices perform similarly, but the latter is easier to implement. So we will report results mainly based on the second realization. In our experiments, we consider two offline RL algorithms under this category, *i.e.*, CRR, and IQL. They use two weighting schemes: $f_{\text{CRR}} = \exp\left[\left(Q_\phi(s,a) - \mathbb{E}_{a' \sim \hat{\pi}(a|s)}Q(s,a')\right)/\tau_{\text{CRR}}\right]$ and $f_{\text{IQL}} = \exp\left[(Q_\phi(s,a) - V_\psi(s))/\tau_{\text{IQL}}\right]$, where $\tau$ refers to the temperature parameter and $V_\psi(s)$ is an additional value network parameterized by $\psi$. We defer the details of these two algorithms to Appendix A.

## 4.4 Comparison to Diffusion-QL

Now we are ready to compare our method and Diffusion-QL comprehensively. Though our EDP shares the same policy parametrization as Diffusion-QL, it differs from Diffusion-QL significantly in the training algorithm. As a result, the computational efficiency and generality of diffusion policies have been improved substantially.

**Efficiency** The diffusion policy affects both policy evaluation (Eqn. (6)) and policy improvement (Eqn. (7)). First, calculating $L_{\text{TD}}(\phi)$ in policy evaluation requires drawing the next action from it. Diffusion-QL uses DDPM sampling while EDP employs a DPM-Solver, which can reduce the sampling steps from 1000 to 15, thus accelerating the training. Second, in policy improvement, Diffusion-QL again applies DDPM for sampling. Then, it calculates the loss function based on sampled actions and backpropagates through the sampling process for network update. This means it needs to forward and backward a neural network for $K$ times each training iteration. As a result, Diffusion-QL can only work with small $K$, *e.g.,* $5 \sim 100$. In comparison, our training scheme only passes through the network once an iteration, no matter how big $K$ is. This enables EDP to use a larger $K$ (1000 in our experiments) to train diffusion policy on the more fine-grained scale. The results in Tab. 1 also show a larger $K$ can give better performance.

**Generality** Diffusion-QL can only work with direct policy optimization, which contains only a small portion of algorithms. Moreover, thanks to their flexibility and high performance, the likelihood-based algorithms are preferred for some tasks (*e.g.*, Antmaze). Our method successfully makes diffusion trainable with any RL algorithm.

## 4.5 Controlled Sampling from Diffusion Policies

Traditionally, a continuous policy is represented with a state-conditional Gaussian distribution. During evaluation time, a policy executes deterministically to reduce variance by outputting the distribution mean as an action. However, with diffusion policies, we can only randomly draw a sample from the underlying distribution without access to its statistics. As a result, the sampling process is noisy, and the evaluation is of high variance. We consider the following method to reduce variance.

**Energy-based Action Selection (EAS)** Recall that the goal of (offline) RL is to learn a policy that can maximize the cumulative return or values. Though the policy $\pi_\theta$ is stochastic, the learned $Q_\phi$ provides a deterministic critic for action evaluation. We can sample a few actions randomly, then use $Q_\phi$ for selection among them to eliminate randomness. EAS first samples $N$ actions from $\pi_\theta$ by using any samplers (*i.e.*, DPM-Solver), then sample one of them with weights proportional to $e^{Q(\boldsymbol{s},\boldsymbol{a})}$. This procedure can be understood as sampling from an improved policy $p(\boldsymbol{a}|\boldsymbol{s}) \propto e^{Q(\boldsymbol{s},\boldsymbol{a})}\pi_\theta(\boldsymbol{a}|\boldsymbol{s})$. All results will be reported based on EAS. See Appendix. C.4 for the other two methods.

## 5 Experiments

We conduct extensive experiments on the D4RL benchmark [2] to verify the following assumptions: 1) Our diffusion policy is much more efficient than the previous one regarding training and evaluation costs. 2) Our diffusion policy is a generic policy class that can be learned through direct and likelihood-based policy learning methods. We also provide various ablation studies on the critical components for better understanding.

**Baselines** We evaluate our method on four domains in D4RL, including Gym-locomotion, AntMaze, Adroit, and Kitchen. For each domain, we consider extensive baselines to provide a thorough evaluation. The simplest method is the classic behavior cloning (BC) baseline and 10% BC that performs behavior cloning on the best 10% data. TD3+BC [6] combines off-policy reinforcement

learning algorithms with BC. OneStepRL [1] first conducts policy evaluation to obtain the Q-value of the behavior policy from the offline dataset, then use it for policy improvement. AWAC [25], AWR [28], and CRR [38] improve policy improvement by adding advantage-based weights to policy loss functions. CQL [17] and IQL [15] constrain the policy evaluation process by making conservative Q updates or replacing the max operator with expectile regression. We also consider the Decision Transformer (DT) [4] baseline that maps offline RL as a sequence-to-sequence translation problem.

**Experimental Setup**   We keep the backbone network architecture the same for all tasks and algorithms, which is a 3-layer MLP (hidden size 256) with Mish [23] activation function following Diffusion-QL [37]. For the noise prediction network $\epsilon_\theta(a^k, k; s)$ in diffusion policy, we first encode timestep $k$ with sinusoidal embedding [36], then concatenate it with the noisy action $a^k$ and the conditional state $s$. We use the Adam [13] to optimize both diffusion policy and the Q networks. The models are trained for 2000 epochs on Gym-locomotion and 1000 epochs on the other three domains. Each epoch consists of 1000 iterations of policy updates with batch size 256. For DPM-Solver [20], we use the third-order version and set the model call steps to 15. We reimplement DQL strictly following the official PyTorch code [27] for fair comparison and we refer to DQL (JAX) for all sample efficiency comparisons. We defer the complete list of all hyperparameters to the appendix due to space limits. Throughout this paper, the results are reported by averaging 5 random seeds.

**Evaluation Protocal**   We consider two evaluation metrics in this paper. First, *online model selection* (OMS), proposed by Diffusion-QL [37], selects the best-performing model throughout the whole training process. However, though OMS can reflect an algorithm's capacity, it is cheating, especially when the training procedure is volatile on some of the tasks. Therefore, we propose another metric to focus on the training stability and quality, which is *running average at training* (RAT). RAT calculates the running average of evaluation performance for ten consecutive checkpoints during training and reports the last score as the final performance.

## 5.1 Efficiency and Reproducibility

In this section, we focus on the training and evaluation efficiency of our efficient diffusion policy. We choose the OMS evaluation metric to make a fair comparison with the baseline method Diffusion-QL [37]. We consider four variants of EDP to understand how each component contributes to the high efficiency of our method. 1) EDP is the complete version of our method. It uses the action approximation technique in Eqn. (9) for policy training and uses DPM-Solver for sampling. 2) *EDP w/o DPM* modifies EDP by replacing DPM-Solver with the original DDPM sampling method in Eqn. 4. 3) *EDP w/o AP* removes the action approximation technique. 4) DQL (JAX) is our Jax implementation of Diffusion-QL.

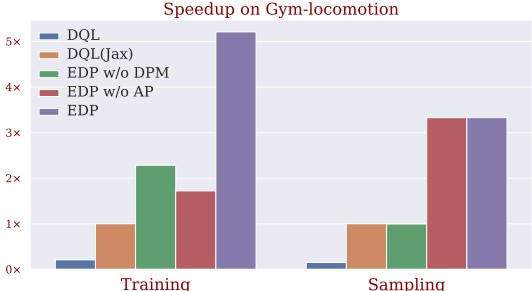

Figure 2: Training and evaluation speed comparison. The training IPS are: 4.66, 22.30, 50.94, 38.4, and 116.21. The sampling SPS are: 18.67, 123.70, 123.06, 411.0, 411.79.

We first benchmark and compare the training/evaluation speed of the above three variants and Diffusion-QL. We choose walker2d-medium-expert-v2 as the testbed. For training speed, we run each algorithm for 10,000 iterations of policy updates and calculate the corresponding iterations-per-second (IPS). Similarly, we sample 10,000 transitions by interacting with the environment and calculate the corresponding steps-per-second (SPS) for evaluation speed. Based on the visualization in Fig. 2, by taking DQL (JAX) as the baseline, we are able to attribute the performance boost to specific techniques proposed. Specifically, we can observe that action approximation makes 2.3x training and 3.3x sampling faster, while using the DPM-Solver adds an additional 2.3x training speedup. We can observe that DQL (JAX) is $5\times$ faster than Diffusion-QL, which means our Jax implementation is more computationally efficient than Diffusion-QL's PyTorch code. This demonstrates that both the action approximation technique and DPM-Solver play a critical role in making the training of diffusion policy efficient. However, this technique does not affect the sampling procedure; thus, *EDP w/o DPM* and DQL (JAX) are on par with each other regarding sampling speed.

Table 2: Average normalized score on the D4RL benchmark. FF denotes the original policies parameterized with feed-forward neural networks, while EDP is our efficient diffusion policy. Results of baselines are taken directly from [15]. For the missing results of 10%, AWAC, and OneStep RL, we re-implement the baselines and report the results. Our results are reported with the RAT metric.

| Dataset | BC | 10%BC | DT | AWAC | OneStep | CQL | TD3+BC | | CRR | | IQL | |
|---|---|---|---|---|---|---|---|---|---|---|---|---|
| | | | | | | | FF | EDP | FF | EDP | FF | EDP |
| halfcheetah-medium-v2 | 42.6 | 42.5 | 42.6 | 43.5 | 48.4 | 44.0 | 48.3 | **52.1** | 44.0 | 49.2 | 47.4 | 48.1 |
| hopper-medium-v2 | 52.9 | 56.9 | 67.6 | 57.0 | 59.6 | 58.5 | 59.3 | **81.9** | 58.5 | 78.7 | 66.3 | 63.1 |
| walker2d-medium-v2 | 75.3 | 75.0 | 74.0 | 72.4 | 81.8 | 72.5 | 83.7 | **86.9** | 72.5 | 82.5 | 78.3 | 85.4 |
| halfcheetah-medium-replay-v2 | 36.6 | 40.6 | 36.6 | 40.5 | 38.1 | 45.5 | 44.6 | **49.4** | 45.5 | 43.5 | 44.2 | 43.8 |
| hopper-medium-replay-v2 | 18.1 | 75.9 | 82.7 | 37.2 | 97.5 | 95.0 | 60.9 | **101.0** | 95.0 | 99.0 | 94.7 | 99.1 |
| walker2d-medium-replay-v2 | 26.0 | 62.5 | 66.6 | 27.0 | 49.5 | 77.2 | 81.8 | **94.9** | 77.2 | 63.3 | 73.9 | 84.0 |
| halfcheetah-medium-expert-v2 | 55.2 | 92.9 | 86.8 | 42.8 | 93.4 | 91.6 | 90.7 | **95.5** | 91.6 | 85.6 | 86.7 | 86.7 |
| hopper-medium-expert-v2 | 52.5 | **110.9** | 107.6 | 55.8 | 103.3 | 105.4 | 98.0 | 97.4 | 105.4 | 92.9 | 91.5 | 99.6 |
| walker2d-medium-expert-v2 | 107.5 | 109.0 | 108.1 | 74.5 | 113.0 | 108.8 | 110.1 | **110.2** | 108.8 | 110.1 | 109.6 | 109.0 |
| average | 51.9 | 74.0 | 74.7 | 50.1 | 76.1 | 77.6 | 75.3 | **85.5** | 77.6 | 78.3 | 77.0 | 79.9 |
| kitchen-complete-v0 | 65.0 | 7.2 | - | 39.3 | 57.0 | 43.8 | 2.2 | 61.5 | 43.8 | 73.9 | 62.5 | **75.5** |
| kitchen-partial-v0 | 38.0 | **66.8** | - | 36.6 | 53.1 | 49.8 | 0.7 | 52.8 | 49.8 | 40.0 | 46.3 | 46.3 |
| kitchen-mixed-v0 | 51.5 | 50.9 | - | 22.0 | 47.6 | 51.0 | 0.0 | **60.8** | 51.0 | 46.1 | 51.0 | 56.5 |
| average | 51.5 | 41.6 | - | 32.6 | 52.6 | 48.2 | 1.0 | 58.4 | 48.2 | 53.3 | 53.3 | **59.4** |
| pen-human-v0 | 63.9 | -2.0 | - | 15.6 | 71.8 | 37.5 | 5.9 | 48.2 | 37.5 | 70.2 | 71.5 | **72.7** |
| pen-cloned-v0 | 37.0 | 0.0 | - | 24.7 | 60.0 | 39.2 | 17.2 | 15.9 | 39.2 | 54.0 | 37.3 | **70.0** |
| average | 50.5 | -1.0 | - | 20.2 | 65.9 | 38.4 | 11.6 | 32.1 | 38.4 | 62.1 | 54.4 | **71.3** |
| antmaze-umaze-v0 | 54.6 | 62.8 | 59.2 | 56.7 | 64.3 | 74.0 | 40.2 | **96.6** | 0.0 | 95.9 | 87.5 | 94.2 |
| antmaze-umaze-diverse-v0 | 45.6 | 50.2 | 53.0 | 49.3 | 60.7 | 84.0 | 58.0 | 69.5 | 41.9 | 15.9 | 62.2 | **79.0** |
| antmaze-medium-play-v0 | 0.0 | 5.4 | 0.0 | 0.0 | 0.3 | 61.2 | 0.2 | 0.0 | 0.0 | 33.5 | 71.2 | **81.8** |
| antmaze-medium-diverse-v0 | 0.0 | 9.8 | 0.0 | 0.7 | 0.0 | 53.7 | 0.0 | 6.4 | 0.0 | 32.7 | 70.0 | **82.3** |
| antmaze-large-play-v0 | 0.0 | 0.0 | 0.0 | 0.0 | 0.0 | 15.8 | 0.0 | 1.6 | 0.0 | 26.0 | 39.6 | **42.3** |
| antmaze-large-diverse-v0 | 0.0 | 6.0 | 0.0 | 1.0 | 0.0 | 14.9 | 0.0 | 4.4 | 0.0 | 58.5 | 47.5 | **60.6** |
| average | 16.7 | 22.4 | 18.7 | 18.0 | 20.9 | 50.6 | 16.4 | 29.8 | 7.0 | 43.8 | 63.0 | **73.4** |

To show that EDP does not hurt performance, we compare the normalized scores of all tasks in Tab. 1. The results for Diffusion-QL are directly copied from the paper, where each task is carefully tuned with its own hyperparameters. Instead, DQL (JAX) and EDP use the same hyperparameters for tasks belonging to the same domain. Moreover, since EDP is more computationally efficient, we can train a better score model with large $K = 1000$, which is larger than the one (5∼100) used in Diffusion-QL. Note that training diffusion policies with large K is impossible in Diffusion-QL, as it needs to forward and backward the neural network K times. Finally, we can observe that EDP and DQL (JAX) are comparable with Diffusion-QL on gym-locomotion tasks but much better on the other three domains. Hence, efficient diffusion policy can boost sample efficiency and improve performance by enabling diffusion training in fine-grained noise prediction.

Table 1: The performance of Diffusion-QL with efficient diffusion policy. The results for Diffusion-QL are directly quoted from [37]. All results are reported based on the OMS metric.

| Dataset | Diffusion-QL | DQL (JAX) | EDP |
|---|---|---|---|
| locomotion | 89.4 | 89.4 | 90.3 |
| antmaze | 75.4 | 78.6 | 77.9 |
| adroit | 68.3 | 66.5 | 89.1 |
| kitchen | 71.6 | 83.0 | 80.5 |

## 5.2 Generality and Overall Results

This section aims to evaluate whether EDP is a general policy class. To this end, we train it with direct policy optimization (TD3) and likelihood-based policy optimization (CRR, IQL) methods. Then, we compare them with their feed-forward counterparts and other baseline methods in Table 2. All scores for diffusion policies are reported using the RAT metric, while other scores are directly quoted from their paper. It shows that EDP can beat the standard Gaussian policy parameterized with an MLP on all domains and for all the three algorithms considered. On the gym-locomotion domain, EDP + TD3 gives the best performance (average score 85.5), while likelihood-based policy learning methods are slightly worse. However, on the other three domains (Kitchen, Adroit, and Antmaze), EDP + IQL beats all the other methods by a large margin (more than 10 average scores). Therefore, we conclude that EDP can serve as a plug-in policy class for different RL methods.

## 5.3 Ablation Study

**Evaluation Metrics** To reveal that evaluation metric is important, we train EDP with TD3 algorithms on three selected environments: walker2d-medium-expert-v2, hopper-medium-expert-v2, and antmaze-medium-diverse-v0. We then compare the scores for OMS (best) and RAT (average) by plotting the training curves in Fig. 3. On

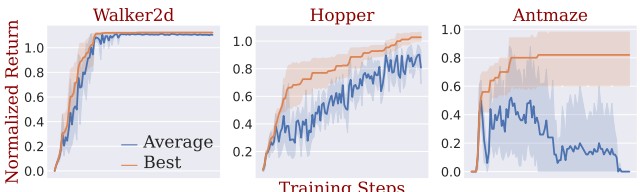

Figure 3: Training curves for EDP +TD3 on three representative environments. Average represents RAT, Best represents OMS.

walker2d, the training is stable; thus, both OMS and RAT scores steadily grow and result in close final scores. A similar trend can be observed on the hopper but with a more significant gap between these two metrics. However, these two metrics diverge significantly when the training succeeds and then crashes on antmaze. Therefore, OMS is misleading and can not give a reliable evaluation of algorithms, which explains the necessity of using RAT in Sec. 5.2.

**Energy-Based Action Selection** We notice that energy-based action selection (EAS) is a general method and can also be used for arbitrary policies. We apply EAS to normal TD3+BC and find no improvement, which shows EAS is only necessary for diffusion sampling. The results are deferred to the Appendix. Moreover, set the number of actions used in

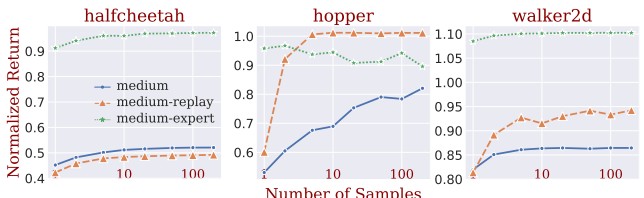

Figure 4: Performance of different number of actions used in EAS. The experiments are conducted on the nine locomotion tasks.

EAS from 1 to 200, and report the performance on gym-locomotions tasks in Fig. 4. It shows the normalized score monotonically grows as the number of actions increases on 8 out of 9 tasks. In our main experiments, we set the number of actions to 10 by trading off the performance and computation efficiency.

**DPM-Solver** We are using DPM-Solver to speed up the sampling process of diffusion policies. The number of models in DPM-Solver is an important hyper-parameter that affects sampling efficiency and quality. We vary this number from 3 to 30 and compare the performance on gym-locomotion tasks in Fig. 5. We can observe that the performance increases as more

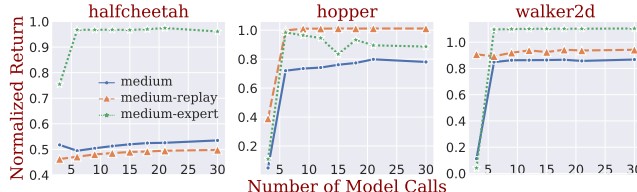

Figure 5: Performance of DPM-Solver with varying steps. The experiments are conducted on the nine locomotion tasks.

steps of model calls are used. The performance gradually plateaus after 15 model calls. Therefore, we use 15 in our main experiments.

## 6  Conclusion

Diffusion policy has emerged as an expressive policy class for offline reinforcement learning. Despite its effectiveness, diffusion policy is limited by two drawbacks, hindering it from wider applications. First, training a diffusion policy requires to forward and backward through a long parameterized Markov chain, which is computationally expensive. Second, the diffusion policy is a restricted policy class that can not work with likelihood-based RL algorithms, which are preferred in many scenarios. We propose efficient diffusion policy (EDP) to address these limitations and make diffusion policies faster, better, and more general. EDP relies on an action approximation to construct actions from corrupted ones, thus avoiding running the Markov chain for action sampling at training. Our benchmarking shows that EDP achieves $25\times$ speedup over Diffusion-QL at training time on the gym-locomotion tasks in D4RL. We conducted extensive experiments by training EDP with various offline RL algorithms, including TD3, CRR, and IQL, the results clearly justify the superiority of diffusion policies over Gaussian policies. As a result, EDP set new state-of-the-art on all four domains in D4RL.

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

## Broader Impact

In this paper, we propose an efficient yet powerful policy class for offline reinforcement learning. We show that this method is superior to most existing methods on simulated robotic tasks. However, some war robots or weapon robots might employ our EDP to learn strategic agents considering the generalization ability of EDP. Depending on the specific application scenarios, it might be harmful to domestic privacy and safety.

## A  Reinforcement Learning Algorithms

In this section, we introduce the details of the RL algorithms experimented.

### A.1  TD3+BC

TD3 [8] is a popular off-policy RL algorithm for continuous control. TD3 improves DDPG [19] by addressing the value overestimation issue. Specifically, TD3 adopts a double Q-learning paradigm that computes the TD-target $\hat{Q}(s, a)$ as

$$\hat{Q}(s, a) = r(s, a) + \gamma \min(Q_1(s', a'), Q_2(s', a')), \quad a' = \pi(s'), \tag{14}$$

where $\pi(s')$ is a deterministic policy, $Q_1$ and $Q_2$ are two independent value networks. Specifically, TD3 takes **only** $Q_1$ for policy improvement

$$\pi^* = \arg\max_\pi \mathbb{E}\left[Q_1(s, \hat{a})\right], \quad \hat{a} = \pi(s). \tag{15}$$

Built on top of TD3, TD3+BC simply adds an additional behavior cloning term for its policy improvement

$$\pi^* = \arg\max_\pi \mathbb{E}_{s,a\sim\mathcal{D}}\left[Q(s, \hat{a}) - \alpha(\hat{a} - a)^2\right], \quad \hat{a} = \pi(s), \tag{16}$$

where $\alpha$ is a hyper-parameter that balances these two terms. However, as the scale of Q-values are different from the behavior cloning loss, TD3+BC normalizes it with $\frac{1}{N}\sum_{i=1}^{N}|Q(s_i, \hat{a}_i)|$ for numerical stability.

In the context of EDP, we observe that 1) the behavior cloning term can be naturally achieved as a diffusion loss as defined in Eqn. 5; 2) the sampled action $\hat{a}$ is replaced by action approximation as in Eqn. 9 for efficient policy improvement. Different from the original TD3+BC that uses only $Q_1$ for policy improvement, we sample from $Q_1$ and $Q_2$ with equal probability for each policy improvement step.

### A.2  Critic Regularized Regression

CRR follows Advantage Weighted Regression (AWR) [29], which is a simple yet effective off-policy RL method, for policy improvement. Specifically, AWR considers a constrained policy improvement step

$$\arg\max_\pi \int_s d_\mu(s) \int_a \pi(a \mid s) A(s, a) da ds, \quad \text{s.t.} \quad \int_s d_\mu(s) D_{\text{KL}}\left[\pi(\cdot \mid s) \,\|\, \mu(\cdot \mid s)\right] \leq \epsilon, \tag{17}$$

where $A(s, a)$ is the advantage function, $\mu$ is a behavior policy that is used to generate trajectories during off-policy RL, $d_\mu(s)$ is the state distribution induced by $\mu$, $D_{\text{KL}}$ is the KL divergence, and $\epsilon$ is a threshold parameter. This constrained optimization problem can be solved in closed form, which gives an optimal policy of

$$\pi^*(a \mid s) = \frac{1}{Z(s)}\mu(a \mid s)\exp\left(\frac{1}{\beta}A(s, a)\right), \tag{18}$$

with $Z(s)$ being the partition function and $\beta$ is a hyper-parameter. For policy improvement, AWR simply distills the one-step improved optimal policy to the learning policy $\pi$ by minimizing the

KL-divergence

$$\arg\min_{\pi} \mathbb{E}_{s \sim d_\mu(s)} \left[ D_{\text{KL}} \left[ \pi^*(\cdot \mid s) \parallel \pi(\cdot \mid s) \right] \right]$$

$$= \arg\max_{\pi} \mathbb{E}_{s \sim d_\mu(s), a \sim \mu(\cdot \mid s)} \left[ \log \pi(a \mid s) \exp(\frac{1}{\beta} A(s, a)) \right] \tag{19}$$

CRR performs policy improvement in the same way as AWR, which simply replaces the sampling distribution with a fixed dataset

$$\arg\max_{\pi} \mathbb{E}_{s, a \sim \mathcal{D}} \left[ \log \pi(a \mid s) \exp(\frac{1}{\beta} A(s, a)) \right]. \tag{20}$$

As a result, this naturally imposes an implicit constraint on its policy improvement step.

However, as computing the log-likelihood $\log \pi(a \mid s)$ is intractable in diffusion models, in practical implementations, we use Eqn. 13 instead. In addition, we compute the advantage by

$$A(s, a) = \min(Q_1(s, a) - Q_2(s, a)) - \frac{1}{N} \sum_{i=1}^{N} \min(Q_1(s, \hat{a}_i), Q_2(s, \hat{a}_i)), \tag{21}$$

where $\hat{a}_i \sim \mathcal{N}(\hat{a}^0, \sigma)$ is a sampled action with the mean of approximated action and an additional standard deviation. In our experiment, we found using a fixed standard deviation, an identity matrix of $\sigma = \mathbb{I}$, $\beta = 1$, and sample size $N = 10$ generally produce a good performance.

### A.3  Implicit Q Learning

Similar to CRR, IQL also adopts the AWR-style policy improvement that naturally imposes a constraint to encourage the learning policy to stay close to the behavior policy that generates the dataset. Different from CRR query novel and potentially out-of-distribution (OOD) actions when computing advantages, IQL aims to completely stay in-distribution with only dataset actions, while maintaining the ability to perform effective multi-step dynamic programming during policy evaluation. IQL achieves this by introducing an additional value function $V(s)$ and performing expectile regression. Specifically, the policy evaluation in IQL is implemented as

$$\min_{V} \mathbb{E}_{s, a \sim \mathcal{D}} \left[ L_2^\tau(Q(s, a)) - V(s) \right]$$

$$\min_{Q} \mathbb{E}_{s, a, s' \sim \mathcal{D}} \left[ (r(s, a) + \gamma V(s') - Q(s, a))^2 \right], \tag{22}$$

where $L_2^\tau$ is the expectile regression loss defined as $L_2^\tau(x) = |\tau - \mathbb{1}(x < 0)|x^2$ with hyper-paramter $\tau \in (0, 1)$. The intuition behind IQL is that with a larger $\tau$, we will be able to better approximate the $\max$ operator. As a result, IQL approximates the Bellman's optimality equation without querying OOD actions.

In practical implementations, we also adopt double Q learning for IQL, where we replace the $Q(s, a)$ in Eqn. 22 with $\min(Q_1(s, a), Q_2(s, a))$, and then use the updated value network to train both Q value networks. For IQL, we follow the policy improvement steps of CRR, as described in Eqn. 20 and Eqn. 21. The key difference is that instead of sampling actions to compute $\frac{1}{N} \sum_{i=1}^{N} \min(Q_1(s, \hat{a}_i), Q_2(s, \hat{a}_i))$, IQL replaces it directly with the learned $V(s)$. As for hyper-parameters, we use a temperature of $\beta = 1$, a fixed standard deviation $\sigma = \mathbb{I}$, and for expectile ratio $\tau$, we use $\tau = 0.9$ for antmaze-v0 environments and $\tau = 0.7$ for other tasks.

## B  Reinforcement Guided Diffusion Policy Details

The overall algorithm for our Reinforcement Guided Diffusion Policy Learning is given in Alg. 1.

The detailed algorithm for energy-based action selection is given in Alg. 2.

## C  Environmental Details

### C.1  Hyper-Parameters

We detail our hyperparameters in Tab. 3.

**Algorithm 1:** Reinforcement Guided Diffusion Policy

---

**Input:** $I, \lambda, \eta, \mathcal{D}, \boldsymbol{\epsilon}_\theta(\boldsymbol{a}^k, k; \boldsymbol{s}), Q_\phi(\boldsymbol{s}, \boldsymbol{a})$
**Output:** $\theta, \phi$
**for** $i = 0, \ldots, I$ **do**
   // Sample a batch of data
   $\{(\boldsymbol{s}, \boldsymbol{a}, \boldsymbol{s}')\} \sim \mathcal{D}$
   // Sample next actions with DPM-Solver
   $\boldsymbol{a}' \sim \pi_\theta(\cdot|\boldsymbol{s})$
   $\phi \leftarrow \phi - \eta\nabla_\phi L_{\text{TD}}(\phi)$
   // Action approximation
   $\hat{\boldsymbol{a}}^0 = \frac{1}{\sqrt{\bar\alpha^k}}\boldsymbol{a}^k - \frac{\sqrt{1-\bar\alpha^k}}{\sqrt{\bar\alpha^k}}\boldsymbol{\epsilon}_\theta(\boldsymbol{a}^k, k; \boldsymbol{s})$
   $\theta \leftarrow \theta - \eta\nabla_\theta(L_{\text{diff}}(\theta) + \lambda L_\pi(\theta))$

---

**Algorithm 2:** Energy-based Action Selection

---

**Input:** number of actions: $N, \pi_\theta(\boldsymbol{a}|\boldsymbol{s}), Q_\phi(\boldsymbol{s}, \boldsymbol{a})$
**Output:** $a$
$\boldsymbol{a}_i \sim \pi_\theta(\boldsymbol{a}|\boldsymbol{s}) \quad i = 1, \ldots, N$
$\boldsymbol{a} = \texttt{categorical\_sample}(\{\boldsymbol{a}_i\}, \{e^{Q_\phi(\boldsymbol{s}, \boldsymbol{a}_i)}\})$

---

## C.2    More Results

We first expand Tab. 1 by providing detailed numbers for each of the tasks used in Tab. 4.

We report the performance of EDP trained with TD3, CRR, and IQL in Tab. 5, where we directly compare the scores of different evaluation metrics, *i.e.*, OMS and RAT. We can observe that there are huge gaps between OMS and RAT for all domains and all algorithms. However, IQL and CRR are relatively more stable than $TD3$. For example, on the antmaze domain, TD3 achieves a best score of 80.5, while the average score is just 29.8. In comparison, the best and average scores of IQL are 89.2 and 73.4, respectively.

## C.3    More Results on EAS

We compare normal TD3+BC, TD3+BC with EAS for evaluation and TD3 + EDP in Tab. 6.

## C.4    More Experiments on Controlled Sampling

As in Sec. 4.5, we discussed reducing variance with EAS. We now detail another two methods experimented as below.

**Policy scaling**    Instead of sampling from the policy directly, we can sample from a sharper policy $\pi_\theta^\tau(\boldsymbol{a}|\boldsymbol{s})$, where $\tau > 1$. The scaled policy $\pi_\theta^\tau$ shares the same distribution modes as $\pi_\theta$ but with reduced variance. Since diffusion policies are modeling the scores $\log \pi_\theta(\boldsymbol{a}_t|\boldsymbol{s})$, policy scaling can be easily achieved by scaling the output of noise-prediction network by the factor $\tau$. We conduct experiments on the gym-locomotion tasks in D4RL by varying $\tau$ from 0.5 to 2.0, as shown in Fig. 6, the results show the best performance is achieved when $\tau = 1.0$. It means sampling from a scaled policy does not work.

**Deterministic Sampling** This method is based on the observation that the sampling process of the DPM-Solver is deterministic except for the first step. The first step is to sample from an isotropic Gaussian distribution. We modify it to use the mean, thus avoiding stochasticity. Consider a initial noise $\boldsymbol{a}^K \sim \mathcal{N}(\boldsymbol{0}, \boldsymbol{I})$, we rescale $\boldsymbol{a}^K$ by a noise scale factor. We show how this factor affects the final performance by varying it from 0.0 to 1.0. As illustrated in Fig. 7, the best performance is achieved at zero noise in most cases, and a normal $\boldsymbol{a}^K$ performs worst. This means reducing the variance of the initial noise is able to improve the performance of a diffusion policy. However, the best performance achieved in this way still falls behind EAS.

Table 3: Hyperparameters used by EDP.

| Task | Learning rate | Gradient norm clipping | Loss weight | Epochs | Batch size |
|---|---|---|---|---|---|
| Locomotion | 0.0003 | 5 | 1.0 | 2000 | 256 |
| Antmaze | 0.0003 | 5 | 1.0 | 1000 | 256 |
| Adroit | 0.00003 | 5 | 0.1 | 1000 | 256 |
| Kitchen | 0.0003 | 5 | 0.005 | 1000 | 256 |

Table 4: The performance of Diffusion-QL with efficient diffusion policy. The results for Diffusion-QL are directly quoted from [37]. EDP is our method. DQL (JAX) is a variant that uses the exact same configurations as Diffusion-QL. All results are reported based on the OMS metric.

| Dataset | Diffusion-QL | DQL (JAX) | EDP |
|---|---|---|---|
| halfcheetah-medium-v2 | 51.5 | 52.3 | 52.8 |
| hopper-medium-v2 | 96.6 | 95.3 | 98.6 |
| walker2d-medium-v2 | 87.3 | 86.9 | 89.6 |
| halfcheetah-medium-replay-v2 | 48.3 | 50.3 | 50.4 |
| hopper-medium-replay-v2 | 102.0 | 101.8 | 102.7 |
| walker2d-medium-replay-v2 | 98.0 | 96.3 | 97.7 |
| halfcheetah-medium-expert-v2 | 97.2 | 97.3 | 97.1 |
| hopper-medium-expert-v2 | 112.3 | 113.1 | 112.0 |
| walker2d-medium-expert-v2 | 111.2 | 111.5 | 112.0 |
| average | 89.4 | 89.4 | 90.3 |
| antmaze-umaze-v0 | 96.0 | 93.4 | 93.4 |
| antmaze-umaze-diverse-v0 | 84.0 | 74.0 | 66.0 |
| antmaze-medium-play-v0 | 79.8 | 96.0 | 88.0 |
| antmaze-medium-diverse-v0 | 82.0 | 82.0 | 96.0 |
| antmaze-large-play-v0 | 49.0 | 66.0 | 60.0 |
| antmaze-large-diverse-v0 | 61.7 | 60.0 | 64.0 |
| average | 75.4 | 78.6 | 77.9 |
| pen-human-v1 | 75.7 | 74.0 | 98.3 |
| pen-cloned-v1 | 60.8 | 59.1 | 79.9 |
| average | 68.3 | 66.5 | 89.1 |
| kitchen-complete-v0 | 84.5 | 100.0 | 97.0 |
| kitchen-partial-v0 | 63.7 | 73.0 | 71.5 |
| kitchen-mixed-v0 | 66.6 | 76.0 | 73.0 |
| average | 71.6 | 83.0 | 80.5 |

## C.5  Computational Cost Comparison between EDP and Feed-Forward Policy networks

We benchmark the training speed of TD3+BC with EDP on walker2d-medium-expert-v2, by training each agent for 10,000 iterations of policy updates. The training speed for TD3+BC is 689 iterations-per-second (IPS), while EDP is 412 IPS. In other words, it takes around 3 hours to train an agent with the feed-forward network, while EDP needs around 5 hours. We also conducted experiments by double the network used in TD3+BC; unfortunately, there was no performance gain on the locomotion tasks (75.3 to 75.6). Moreover, both feed-forward policy and diffusion policy utilize a 3-layer MLP (hidden size 256) as the backbone network. Therefore, the network capacity should be comparable.

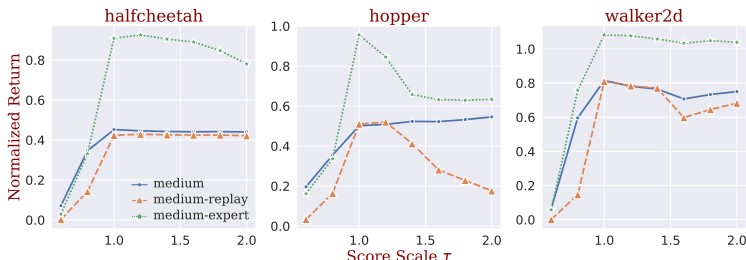

Figure 6: Performance of EDP + TD3 on gym-locomotion tasks with varying $\tau$.

Table 5: Average normalized score on the D4RL benchmark of EDP trained with different algorithms. "Best" represents the online model selection metric, while "Average" is our runing average at training metric.

| | EDP + TD3 | | EDP + CRR | | EDP + IQL | |
|---|---|---|---|---|---|---|
| | Average | Best | Average | Best | Average | Best |
| halfcheetah-medium-v2 | 52.1 | 52.8 | 49.2 | 50.2 | 48.1 | 48.7 |
| hopper-medium-v2 | 81.9 | 98.6 | 78.7 | 95.0 | 63.1 | 97.3 |
| walker2d-medium-v2 | 86.9 | 89.6 | 82.5 | 85.8 | 85.4 | 88.7 |
| halfcheetah-medium-replay-v2 | 49.4 | 50.4 | 43.5 | 47.8 | 43.8 | 45.5 |
| hopper-medium-replay-v2 | 101.0 | 102.7 | 99.0 | 101.7 | 99.1 | 100.9 |
| walker2d-medium-replay-v2 | 94.9 | 97.7 | 63.3 | 89.8 | 84.0 | 93.4 |
| halfcheetah-medium-expert-v2 | 95.5 | 97.1 | 85.6 | 93.5 | 86.7 | 80.9 |
| hopper-medium-expert-v2 | 97.4 | 112.0 | 92.9 | 109.4 | 99.6 | 95.7 |
| walker2d-medium-expert-v2 | 110.2 | 112.0 | 110.1 | 112.3 | 109.0 | 111.5 |
| average | 85.5 | 90.3 | 78.3 | 87.3 | 79.9 | 84.7 |
| kitchen-complete-v0 | 61.5 | 93.4 | 73.9 | 95.8 | 75.5 | 95.0 |
| kitchen-partial-v0 | 52.8 | 66.0 | 40.0 | 56.7 | 46.3 | 72.5 |
| kitchen-mixed-v0 | 60.8 | 88.0 | 46.1 | 59.2 | 56.5 | 70.0 |
| average | 58.4 | 96.0 | 53.3 | 70.6 | 59.4 | 79.2 |
| pen-human-v0 | 48.2 | 60.0 | 70.2 | 127.8 | 72.7 | 130.3 |
| pen-cloned-v0 | 15.9 | 64.0 | 54.0 | 106.0 | 70.0 | 138.2 |
| average | 32.1 | 77.9 | 62.1 | 116.9 | 71.3 | 134.3 |
| antmaze-umaze-v0 | 96.6 | 98.3 | 95.9 | 98.0 | 94.2 | 98.0 |
| antmaze-umaze-diverse-v0 | 69.5 | 79.9 | 15.9 | 80.0 | 79.0 | 90.0 |
| antmaze-medium-play-v0 | 0.0 | 89.1 | 33.5 | 82.0 | 81.8 | 89.0 |
| antmaze-medium-diverse-v0 | 6.4 | 97.0 | 32.7 | 72.0 | 82.3 | 88.0 |
| antmaze-large-play-v0 | 1.6 | 71.5 | 26.0 | 57.0 | 42.3 | 52.0 |
| antmaze-large-diverse-v0 | 4.4 | 73.0 | 58.5 | 71.0 | 60.6 | 68.0 |
| average | 29.8 | 80.5 | 43.8 | 76.7 | 73.4 | 89.2 |

Table 6: Energy-based Action Selection + Normal TD3

| Dataset | TD3+Diff | TD3+EAS | TD3 |
|---|---|---|---|
| halfcheetah-medium-v2 | 52.1 | 48.7 | 47.8 |
| hopper-medium-v2 | 81.9 | 50.8 | 54.0 |
| walker2d-medium-v2 | 86.9 | 77.7 | 53.4 |
| halfcheetah-medium-replay-v2 | 49.4 | 44.4 | 44.1 |
| hopper-medium-replay-v2 | 101.0 | 59.8 | 59.1 |
| walker2d-medium-replay-v2 | 94.9 | 74.8 | 71.6 |
| halfcheetah-medium-expert-v2 | 95.5 | 85.9 | 92.3 |
| hopper-medium-expert-v2 | 97.4 | 71.9 | 95.1 |
| walker2d-medium-expert-v2 | 110.2 | 108.4 | 110.0 |
| gym-locomotion-v2 (avg) | 85.5 | 69.2 | 69.7 |

## C.6 The effect of action approximation

We compare with and without action approximation on the following three environments by using OMS metric (Tab. 4). In Tab. 7, the DDPM column will forward and backward a policy network 100 times at training time, but action approximation only needs once. We can observe that action approximation will slightly harm the performance, when the same number of diffusion steps is used. However, it supports training diffusion policies with larger K (e.g. 1000), while Diffusion-QL does

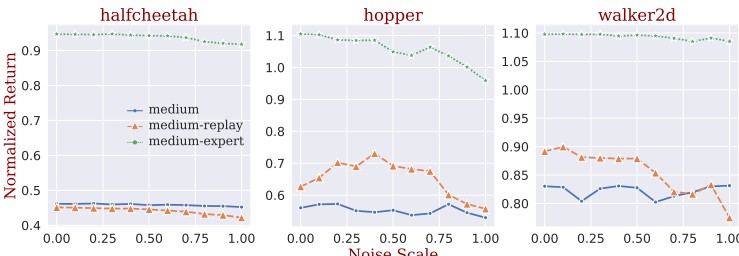

Figure 7: Performance of EDP + TD3 on gym-locomotion tasks with varying initial noise scale.

Table 7: The effect of action approximation.

| OMS K=100 | DDPM | Action Approx |
|---|---|---|
| walker2d-medium-v2 | 86.9 | 85.5 |
| walker2d-medium-reply-v2 | 96.3 | 93.3 |
| walker2d-medium-expert-v2 | 111.5 | 111.1 |

not. Increasing K is able to avoid performance drop as evidenced by the last column of Tab. 2 and Tab. 4.

## D Negative Societal Impacts

In this paper, we propose an efficient yet powerful policy class for offline reinforcement learning. We show that this method is superior to most existing methods on simulated robotic tasks. However, some war robots or weapon robots might employ our EDP to learn strategic agents considering the generalization ability of EDP. Depending on the specific application scenarios, it might be harmful to domestic privacy and safety.

