# OpenReview forum: "Efficient Diffusion Policies For Offline Reinforcement Learning"
_NeurIPS.cc/2023/Conference — NeurIPS 2023 poster_

### Official Review · Reviewer_UWuP · 2023-07-04

**Soundness:** 3 good
**Presentation:** 3 good
**Contribution:** 2 fair
**Rating:** 5
**Confidence:** 4

**Summary:**

This paper focuses on the improvement of computation efficiency of Diffusion-QL by adopting the property of marginal distribution in the diffusion model and the variance control scheme proposed by DPM-Solver. Besides, this paper extends the scope of compatibility with other offline RL methods, from value-based to policy gradient methods. This is achieved by directly approximating the clean examples from corrupted examples at arbitrary diffusion time steps.

**Strengths:**

- This paper is well-written, and the proposed method is easy to understand.
- The evaluation of value at risk seems interesting.
- The technique seems sound, and the results seem strong.


**Weaknesses:**

- The discussion of limitation seems to be neglected.
- More clear discussion is needed to highlight the contribution of this paper and show the novelty.
- The proposed algorithm seems to significantly increase the inference computation complexity by introducing 10000 iterations for each action.
- More experiments are needed to support the contributions sufficiently.
- Some typos are needed to fix through further proofreading.


**Questions:**

See my detailed comments below.

==Major concerns==

- In lines 233-235 of Section 4.5: according to Equation (9), the action approximation is the mean value of action Gaussian distribution, so why adopting $\hat{a}^0$ can not reduce high variance?
- In Section 4.4, the improvement of sample efficiency (from $ K=1000$ to $K\approx 15$) comes from DPM-Solver rather than the action approximation. Firstly, if the author adopts action approximation, whether the $a_K$ comes from normal Gaussian distribution? If so, the noise prediction function $\epsilon_theta (a^K,K;s)$ should predict the action based on the state and the Gaussian noise $a^K$. Intuitively, generating proper actions based on states seems more difficult, i.e., that classical policy function.
- As for the log-likelihood, why can we not obtain the exact log-likelihood of $\pi_{\theta}(a|s)$ through $log~p(a^K)+\sum_{k=1}^{K}log~p_{\theta}(a^{k-1}| a^{k}, s)$?
- Does the action approximation improve the computation efficiency of diffusion policy?
- I hope the authors explain how they use the action approximation during training.
- During the sampling stage, the DPM-Solver speeds up generation rather than action approximation. If action approximation just works on Equation (10), I think it is very similar to the “Target Policy Smoothing Regularization” technique in Section 5.3 of the TD3 paper. I want to know why the authors use the predicted $\hat{a}^0$ rather than $a^0+\epsilon$.


==Minor concerns==
- How does the author avoid out-of-distribution actions' effects on the Q function?
- DPM-Solver directly makes me understand the sample efficiency of EDP, but I can not understand why the training efficiency can also be improved? So I hope more description of the reason.
- In line 318-320 of Section 5.1: during training the diffusion model, we sample a step $t\sim [1, K]$, then train the diffusion model according to Equation (5). If the $K$ is large, we can still train the diffusion model. So I hope the authors explain this claim more clearly.


==Typos==
- Line 188: Eqn. 7 -> Eqn. (7).
- Line 176: Eqn. 1 -> Eqn. (1).
- Line 201: Eqn. 11 -> Eqn. (11).
- Line 292: Eqn. 4 -> Eqn. (4).
- Line 518 of Appendix: Eqn. 5 -> Eqn. (5).
- Line 519 of Appendix: Eqn. 9 -> Eqn. (9).
Similar typos can also be found in many places in Appendix.
- Line 565 of Appendix: Tab. ??.
- Line 575 of Appendix: Tab. ??.


**Limitations:**

The authors do not discuss the limitations and discuss broader impacts.

---

> ### Author Rebuttal · Authors · 2023-08-09
>
> > Q1. In lines 233-235 of Section 4.5, why adopting $\hat{a}^0$ can not reduce high variance?
>
> A1. Here is an intuitive explanation. Given an actual action $a$, action approximation $\hat{a}^0$ represents the mean of the action Gaussian distribution that can be denoised from $a^k$. Therefore, it is related to a specific example drawn from a multi-modal distribution.   $\hat{a}^0$ can be used to guide the optimization gradient for sample $a$, but can not represent the whole data distribution.
>
> > Q2. The improvement of sample efficiency comes from DPM-Solver rather than action approximation.
>
> A2. Yes, the sample or inference efficiency solely comes from DPM-Solver. However, action approximation can greatly boost the training or policy improvement efficiency, as discussed in Sec. 4.2 and Sec. 4.4. This is because one needs to backpropagate the whole sampling chain of an action to optimize the policy. Instead of sampling an action through multiple steps of denoising diffusion, action approximation just needs to forward the network once.
>
> > Q3. Whether the $a^K$ comes from a normal Gaussian distribution?
>
> A3. Yes, when $k=K$, $a^k$ is drawn from a normal gaussian distribution. However, $k$ is uniformly sampled from the discrete set ${1,2,..,K}$, which means $k=K$ is only a small portion.
>
> > Q4. As for the log-likelihood, why can we not obtain the exact log-likelihood of
>  through  $\log p(a^K)+\sum_{k=1}^{K} \log p_{\theta}(a^{k-1}| a^{k}, s) $
>
> A4. This is because $\pi_\theta$ is actually written as the following form,
> $$\pi_\theta(a|s) = \int_{a^K}\int_{a^{K-1}}\dots\int_{a^1} p(a^K|s) p_\theta (a^{K-1}|a^K, s) \dots p_\theta(a^0|a^1,s) d_{a^K} d_{a^{K-1}}\dots d_{a^1}$$
> where the integrations are intractable.
>
> > Q5. Does the action approximation improve the computation efficiency of diffusion policy?  I hope the authors explain how they use the action approximation during training.
>
> A5. Sure, action approximation can boost the training efficiency greatly. Recall the policy optimization procedure in TD3. One needs to first draw a sample from the policy, then feed it into the Q-network to get a Q estimation. Then, the policy is optimized to maximize the Q estimation. This procedure requires backpropagation through the action sampling process. Without action approximation, It takes K times to query the diffusion network for action sampling, thus K times of backpropagation. However, with action propagation, we just need to forward the denoising diffusion network once, which greatly saves GPU memory and optimization time.
>
> > Q6. I want to know why the authors use the predicted $\hat{a}^0$, rather than $a^0 + \epsilon$.
>
> A6. This is because the Q-network takes a clean example as input. Instead, $a^0 + \epsilon$ is a noisy one, feeding it into the Q-network can not give a proper estimation of the corresponding Q value.
>
> > Q7. How does the author avoid out-of-distribution actions' effects on the Q function?
>
> A7. Apart from the RL objective, we also have a behavior cloning term (i.e., the diffusion objective), which restricts the policy to be similar with the behavior policy.
>
> > Q8.  why the training efficiency can also be improved?
>
> A8. Please refer to A5. More discussion on this problem is welcomed.
>
> > Q9. Why can we not train Diffusion-QL when K is large, but train diffusion model?
>
> A9. The difference is that Diffusion-QL needs to sample an example and backpropogate through its sampling chain at training time, but normal diffusion models do not. When K is large, the sampling chain will involve thousands of neural networks, resulting in a super huge computation graph. Based on our empirical experience, it is even computational intractable to compile such a big graph with JAX. Feel free to try with our code by setting K=1000, and disable action approximation.

---

> > ### Comment · Reviewer_UWuP · 2023-08-13
> >
> > I appreciate the authors' responses that clarified my questions. After reviewing their explanations, this paper possesses the necessary qualities for acceptance.

---

> > > ### Author Response · Authors · 2023-08-15
> > >
> > > Thank you so much for your positive feedback and for acknowledging the merit of our work. We truly appreciate the time and effort you have dedicated to comprehensively reviewing our manuscript. For all the typos raised by you, we will definitely fix all of them the future version.
> > >
> > > We would be extremely grateful if you could kindly update your rating.

---

### Official Review · Reviewer_SXfY · 2023-07-05

**Soundness:** 3 good
**Presentation:** 3 good
**Contribution:** 4 excellent
**Rating:** 7
**Confidence:** 3

**Summary:**

The authors propose a method to efficiently train diffusion based policies in the offline-RL setting. The authors suggest three main tricks to enable this: 1) Removing the need to backpropagate through the diffusion sampling chain to update the policy by using what the authors call action approximation; 2) Replacement of intractable policy log likelihoods in reinforcement learning objectives with the ELBO, which allows using diffusion policies generally with many popular offline RL algorithms; 3) Using a fast DPM-Solver for action sampling during policy execution. Altogether, these tricks allow for relatively fast training of expressive multimodal policies which achieve state of the art results across the D4RL benchmark suite.

**Strengths:**

* The paper is generally well written and easy to follow.
* The use of action approximation facilitates training diffusion policy models with much larger diffusion noising time K, with the authors using K=1000. The RL community has recently taken great interest in the prospect of using the expressiveness of diffusion models for policies, and this simple trick seems to make diffusion policy training practical without any noticeable drawbacks.
* The authors also demonstrate that we can use diffusion policies with other common offline RL algorithms such as TD3+BC, CRR and IQL by simply replacing the log likelihood with the diffusion ELBO. This is a valuable contribution to the field, as it could open the door to use diffusion policies to work with many general RL algorithms, a strict improvement over simple gaussian policies.
* The authors show strong empirical performance across the D4RL benchmark suite. EDP policies are especially strong compared to the FF counterparts on the more challenging antmaze and kitchen suite, which involve high multimodality due to undirected demonstrations.

**Weaknesses:**

* The main proposal to efficiently train diffusion policies hinges on action approximation, which uses a much higher variance estimate of $\hat{a}^0$. There are no ablations to show how this affects the final diffusion policy as compared to one trained without action approximation. There is a comparison against Diffusion-QL in the paper, but with a very large difference in diffusion timesteps K, so the effect of action approximation itself is difficult to gauge. I have listed some questions related to this point below, which if clarified would be helpful.
* The use of DPM-Solver is highlighted as a major contribution is speeding up the sampling process compared to DDPM sampling in Diffusion-QL. It is a well known fact that faster diffusion sampling methods than DDPM exist which can be used for sampling, and Diffusion-QL could have used this as well. While it is useful that the authors show the time savings for using DPM-Solver over DDPM sampling, this is not a novel contribution.
* There isn’t discussion about important hyperparameters related to the method. A very important hyperparameter that could be discussed more the diffusion timesteps K (I have noted this as a question below as well). Are EDPs more brittle to train than their FF counterparts in algorithms like IQL with regards to changing hyperparameters?

**Questions:**

* Generally, diffusion models generate higher quality samples with larger diffusion timesteps K, and the authors note in Tab. (1) that training EDP with K=1000 matched or beat Diffusion-QL which was trained with much smaller K~5 to 100. The authors cite this ability to train diffusion models with higher K without having to backprop through the chain as one of the main advantages of EDP. I am curious how much of an impact K actually has in these offline RL benchmark tasks, given that Diffusion-QL has similar scores. I am interested to see an ablation study (at least in some environments) where EDP is trained with different values of K starting at K=1, to see if generally increasing K actually significantly improves policies in these domains.
* Energy-based Action Selection (EAS) seems necessary to come close to the performance of DQL using the proposed EDP policies. The appendix shows that other ways to sample actions from the diffusion model result in much worse performance. DQL does not to my knowledge require EAS to produce good action samples from its diffusion policy. Is this due to the effect of action approximation resulting in training a worse diffusion model?
* As an extension to the above point, have the authors considered using sampling guidance techniques like Classifer-Free Guidance (CFG) to reduce variance of the final policy?
* The authors allow for training diffusion policies with popular offline RL algorithms which require policy log likelihoods by replacing it with the ELBO. Could this be done with lower bounds of simpler deep generative models, such as the ELBO of a Variational Autoencoder as well? I am not aware of prior work that has done this. It could be that diffusion policies are overkill for the expressiveness needed to learn high value policies in the tasks under consideration, in which case these same tricks could be applied to train simpler VAE policies for example.

**Limitations:**

The authors have not addressed technical limitations of their work. They have briefly addressed societal impact of better reinforcement learning methods in the appendix.

---

> ### Author Rebuttal · Authors · 2023-08-09
>
> > Q1. The effect of action approximation
>
> A1. We compare with and without action approximation on the following three environments by using OMS metric (Table. 4). In the following table, the DDPM column will forward and backward a policy network 100 times at training time, but action approximation only needs once. We can observe that action approximation will slightly harm the performance, when the same number of diffusion steps is used. However, it supports training diffusion policies with larger K (e.g. 1000), while Diffusion-QL does not. Increasing K is able to avoid performance drop as evidenced by the last column of Table 1 and Table 4.
>
>
> |         OMS K=100         |  DDPM | Action Approx |
> |:-------------------------:|:-----:|:-------------:|
> |     walker2d-medium-v2    |  86.9 |      85.5     |
> |  walker2d-medium-reply-v2 |  96.3 |      93.3     |
> | walker2d-medium-expert-v2 | 111.5 |     111.1     |
>
> > Q2. Using DPM-Solver is not a novel contribution.
>
> A2. We totally agree that simply replacing the DDPM sampler with DPM-Solver is not a novel contribution. We will modify the statement accordingly to highlight that DPM-Solver is able to improve efficiency but not novelly developed by us. A diffusion policy that can be efficiently trained and a generation diffusion policy compatible with likelihood-based methods are the main contributions.
>
> > Q3. Are EDPs more brittle to train than their FF counterparts in algorithms like IQL with regards to changing hyperparameters?
>
> A3. In our experiments, we do not tune the IQL parameters when it is combined with EDP. For the hyper-parameters for the diffusion part, we just adopt the one used in TD3+EDP. Therefore, we believe that EDPs are quite stable to train with IQL.
>
> > Q4.  if generally increasing K actually significantly improves policies in these domains.
>
> A4. As discussed in Q1, when K is small, action approximation generally performs worse than its full-chain counterparts. Moreover, we analyze the effect of K with EDP on three gym-locomotion tasks. We can conclude that increasing K is able to improve the performance steadily. EDP often fails when K is smaller than 10, but becomes steady when K is larger than 50.
>
>
> |           EDP  K          |   1  |   2  |   3  |   5   |   10  |   50   |   100  |
> |:-------------------------:|:----:|:----:|:----:|:-----:|:-----:|:------:|:------:|
> |     walker2d-medium-v2    | 1.08 | 4.65 | 5.78 | 10.05 | 81.22 |  84.08 |  85.50 |
> |  walker2d-medium-reply-v2 | 0.45 | 1.76 | 3.10 |  4.01 | 18.89 |  89.18 |  93.03 |
> | walker2d-medium-expert-v2 | 0.02 | 2.00 | 4.73 |  8.34 | 80.92 | 109.90 | 111.00 |
>
> > Q5. Is the usage of EAS due to the effect of action approximation resulting in training a worse diffusion model?
>
> A5. This is a really great question. Actually Diffusion-QL suffers the same high-variance problem. And their code actually involves a similar technique like EAS, which is not discussed in their paper. Please refer to their official `diffusion-rl` repo belonging to organization `Twitter` for details. The code snippet is located at line 169-176 of file `agents/ql_diffusion.py`.
>
> > Q6. Using Classifier-free guidance (CFG) to reduce variance of the final policy.
>
> A6. That is a really great suggestion. Unfortunately, we did not try this before mainly due to the following reason. First, It is hard to decide what conditional information to use, probably we can use normalized return as a condition. Second, we need to retrain a diffusion policy such that it can take a conditional or an idle signal as input. Instead, EAS can serve as a plug-in component for evaluation without modifying the training procedure.
>
> > Q7. Can simpler deep generative models be used as policies?
>
> A7. Sure, the answer is yes. BCQ [R1] has successfully trained a conditional-VAE as the diffusion policy. Diffusion-QL also studies different forms of generative models as policies and compares them with diffusion models.  The conclusion is that Diffusion policies show superior ability at capturing data with strong multi-modalities. So, when the data distribution is not so complicated, simpler generative models are definitely a promising choice.
>
> [R1] Fujimoto, Scott, David Meger, and Doina Precup. "Off-policy deep reinforcement learning without exploration." International conference on machine learning. PMLR, 2019.

---

> > ### Comment · Reviewer_SXfY · 2023-08-16
> >
> > Thank you for the detailed response to my questions. I will maintain my score recommending acceptance of the paper.

---

> > > ### Author Response · Authors · 2023-08-17
> > > **Thank you for your feedback**
> > >
> > > We appreciate your detailed review and suggestions. We will incorporate the comments and involve new results into the final revision. Thank you again!

---

### Official Review · Reviewer_44Ux · 2023-07-06

**Soundness:** 3 good
**Presentation:** 3 good
**Contribution:** 3 good
**Rating:** 6
**Confidence:** 4

**Summary:**

The paper proposes to learn a policy for several offline RL tasks in the D4RL benchmark by parameterizing a policy with a diffusion model. The authors claim that their approach is computationally efficient and more compatible with several other RL approaches like maximum likelihood based approaches when compared to Diffusion-QL which is the primary baseline in their experimental studies.

**Strengths:**

1) The paper is well written and easy to read
2) The authors give a clear and concise preface to the method and results in the introduction part of the paper which sets the flow for the rest of the paper.
3) The authors compare their approach with the Diffusion-QL method which introduced the idea of formulating policies as a diffusion model. They explain the issues with the Diffusion-QL approach and try to overcome those shortcomings through EDP
4) Their approach is compatible with max-likelihood based approaches such as IQL (which is one of the SoTA in Offline RL)
5) The run extensive experiments and ablations on the D4RL benchmark and the results reported are consistently better than Diffusion-QL


**Weaknesses:**

1) Line 206: The authors claim that both policy approximations are empirically similar when calculating the objective though there is no evidence provided for the same. Would be good to know their intuition on that as well.
2) There is no discussion around why EDP + IQL works significantly better in some tasks like kitchen, adroit whereas it under-performs in the locomotion tasks (where EDP + TD3) performs better. Would be good to know why one works better than another in these cases. Is there any investigation done in that direction?
3) Barely no discussion around limitations of the approach.
4) Would the approach be robust to data from multi-modalities? There is a mention of why that would be an issue with gaussian policies but there is no details on that in the latter parts of the paper.
5) Would the method work on more complex offline-RL task like CARLA? (why was that excluded from the investigation? I understand that diffusion-QL does not test their method on this task but was it ever tried out?)


**Questions:**

Please refer to some of the questions mentioned in the weakness section;

Minor comments:

1) Is the speed only reason for using the DPM-Solver? Are there other alternatives that were considered

2) Please rephrase line 274 (Do not like phrasing the evaluation protocol of another peer-reviewed paper to be labelled as cheating)

**Limitations:**

I do not see any discussion around limitations of the approach or other assumptions made.

---

> ### Author Rebuttal · Authors · 2023-08-09
>
> > Q1. Eqn. (12) and Eqn. (13) are empirically similar to each other. No evidence
>
> A1. We ran experiments with Eqn. (13) on three environments with TD3 as the base algorithm. These two approximations are compared in the following table. We can observe that they indeed perform similarly.
>
> |    Base algorithm: TD3    | Eqn. (12) | Eqn. (13) |
> |:-------------------------:|:---------:|:---------:|
> |     walker2d-medium-v2    |    86.9   |    86.9   |
> |  walker2d-medium-reply-v2 |    94.8   |    94.9   |
> | walker2d-medium-expert-v2 |   110.3   |   110.2   |
>
> > Q2. Why EDP-IQL out-performs in kitchen, adroit and antmaze, but underperforms in locomotion.
>
> A2. Thank you for raising this valuable question. EDP-IQL actually shares the same observation as IQL. But, why does this phenomenon happen to IQL? Our hypothesis is that compared to locomotion tasks, the other three types of tasks have sparser reward. As a result, the out-of-distribution issue is amplified in these tasks. IQL performs in-distribution value estimation without querying any out-of-distribution actions, making it less prone to overestimation in such sparse tasks, resulting in better performance.
>
> > Q3. Barely no discussion around limitations of the approach.
>
> A3. Thank you very much for pointing out this problem. We will add the following discussion in the revision.
>
> Though EDP is a much more efficient class of diffusion policies, it is still inefficient compared to a feedforward policy network. For example, the training time on walker2d for a feedforward policy network is just 2 hours, while EDP takes 5 hours. Moreover, The implementation of diffusion policies is much more complicated than its feedforward counterparts.
>
> > Q4. Would the approach be robust to data from multi-modalities?
>
> A4. Sure, diffusion models/policies are superior in modeling multimodal data. We do not provide detailed experimental analysis on this as Diffusion-QL has already conducted a comprehensive study. If you are interested, please refer to Figure 1 of Diffusion-QL for more details.
>
> > Q5. Would the method work on more complex offline-RL task like CARLA?
>
> A5. Sorry, we just tested our methods following conversions. Extending diffusion policies to visual/pixel-based environments itself is a research topic.
>
> Thank you for the valuable suggestion! We agree that CARLA is definitely an interesting domain for testing the capability of EDP on more complex tasks. However, in this paper, we only focus on prototyping a general diffusion policy for offline RL, handling the vision inputs of CARLA and imbalanced distribution in self-driving scenarios is out of the scope of this paper. We will leave it for future study.
>
> > Q6. Is the speed only reason for using the DPM-Solver? Are there other alternatives that were considered
>
> A6. Yes, that is the only reason. Sure, we actually tried DDIM, it performs similar to DPM-Solver, but a bit slower. Therefore, we only report DPM-Solver in our paper.
>
> > Q7. Please rephrase line 274 (Do not like phrasing the evaluation protocol of another peer-reviewed paper to be labelled as cheating)
>
> A7. Definitely, we will remove the “cheating” statement and focus on the difference between evaluation protocols only.

---

> > ### Comment · Reviewer_44Ux · 2023-08-16
> >
> > Thank you for the answers to my questions/concerns. I believe most of my questions were answered comprehensively. I hope you make the necessary changes as you have mentioned in the above rebuttal. I will maintain my score to acceptance with these changes being included.

---

> > > ### Author Response · Authors · 2023-08-17
> > > **Thank you for your feedback**
> > >
> > > We appreciate your detailed comments and suggestions. We will polish our paper further and incorporate new changes into the final revision. Thank you again!

---

### Official Review · Reviewer_TjdE · 2023-07-06

**Soundness:** 3 good
**Presentation:** 3 good
**Contribution:** 3 good
**Rating:** 5
**Confidence:** 4

**Summary:**

The focus of this paper is to enhance the diffusion policies introduced in Diffusion-QL for offline reinforcement learning. The authors address the challenges of training and sampling efficiency by incorporating action approximation and employing an advanced ODE solver for diffusion policies. They conducted extensive experiments to demonstrate the effectiveness of their proposed approach, known as Efficient Diffusion Policy (EDP).

**Strengths:**

1.	This paper is well structured and easy to follow.
2.	The conducted experiments are of enough quantity and quality.



**Weaknesses:**

> I think some claims need to be more careful or accurate.

“… reduce the diffusion policy training time from 5 days …”
“However, it takes tens to hundreds more time to train a diffusion policy than a diagonal Gaussian one”

I tried Diffusion-QL before and I don’t think it takes about 5 days for training in their default setting (K=5). In my case, it is only about 20 hours, so I am not sure what setting is inferred here to be 5 days.

Moreover, “In comparison, our training scheme only passes through the network once an iteration, no matter how big K is.” This statement is in correct. According to Algorithm 1 and the description in the paper, “To improve the efficiency of policy evaluation, we propose to replace the DDPM sampling in Eqn. (4) with DPM-Solver”, the policy evaluation part still needs the action samples with the full reverse process, which takes a number of steps even with DPM-Solver.

> The novelty is kind of limited. In short words, EDP tried two things: 1. replace the real action samples generated by the full reverse process in policy improvement step with one-time real action prediction. 2. replace the original DDPM solver with DPM-solver.

> The Energy-based Action Selection (EAS) is not a new thing and has been studied by other works, such as “Offline Reinforcement Learning via High-Fidelity Generative Behavior Modeling (ICLR 2023)”.

> Following diffusion model conventions, I use $x$ to represent $a$ here. This paper uses $\hat{x_{0, t}}$ to replace $x_0$ in policy improvement step. Note using $\hat{x_{0, t}}$ to replace $x_0$ is not a common thing in diffusion models, since the $\hat{x_{0, t}}$ has different meaning compared to $x_0$. $\hat{x_{0, t}}$ usually represents the mean of all $\hat{x_0}$ that could goes back to the true $x_0$ distribution from time step $t$. Hence, $\hat{x_{0, t}}$ could be with much noise and averaged when $t$ is large. Not sure why here the $\hat{x_{0, t}}$ could replace the $x_0$ in Eq (7). If so, why this procedure could not be applied onto Eq (6) as well?



**Questions:**

See weakness.

**Limitations:**

Yes

---

> ### Author Rebuttal · Authors · 2023-08-09
>
> > Q1. About the training time of Diffusion-QL.
>
> A1. Thank you for bringing this valuable problem into discussion. We noticed that in Diffusion-QL’s official code repo, they default the number of diffusion steps to 100 (K=100). Please refer to their official `diffusion-rl` repo belonging to organization `Twitter` for details. The hyper-parameter is located at line 187 of file `run_offline.py`. Therefore, to make sure we can reproduce their results, we stick to this setting throughout the experiments. It indeed takes 5 days to finish one experiment.
>
> > Q2. Incorrect statement about “our training scheme only passes through the network once an iteration, no matter how big K is”.
>
> A2. Sorry for the confusion caused. In line 218-220, we are discussing the policy evaluation efficiency. It says DPM-Solver reduces the number of steps to 15. However, in line 221-225 (where the quoted sentence appears), we are talking about the training efficiency. For training, EDP just needs to forward and backward through the network once per iteration, thanks to the proposed action approximation technique. Please see Sec. 4.2 for more details.
>
> > Q3. limited novelty, EDP tried two things: 1) action approximation, 2) DPM-solver
>
> A3. Thank you for opening this discussion on the novelty. We do agree that our proposed methods are quite simple. We’d like to emphasize that the contribution/novelty of our paper is two-fold. 1) A class of diffusion policy with superior training efficiency, which relies on two techniques, **i.e.** , action approximation and DPM-solver.  2) A more general class of diffusion policy that is compatible with both DDPG-style and likelihood-based RL methods (See Sec.4.3).
>
> > Q4.  EAS has been studied by “Offline Reinforcement Learning via High-Fidelity Generative Behavior Modeling (ICLR 2023)”.
>
> A4. Thank the reviewer for reminding us of this important work. We will discuss this paper in the future version. We’d like to emphasize that our main contribution is proposing an efficient and general diffusion policy, as tested with three different RL algorithms. EAS is a minor contribution for reducing the variance of the learned policy. We agree that the formulation of EAS is not novel, as similar forms have been explored for policy improvement in MPO[R1,R2]. However, EAS is developed for a very specific and different purpose. Moreover, we found EAS differs from SfBC in the following aspects:
> - Motivation: EAS is developed to reduce the variance of the learned diffusion policy. SfBC aims to model a diverse policy.
> - Usage time: EAS is used at evaluation time only. SfBC is used during training time.
> - Role: EAS is for policy execution. SfBC is for value estimation.
>
>
> [R1] Abdolmaleki, Abbas, et al. "Relative entropy regularized policy iteration." arXiv preprint arXiv:1812.02256 (2018).
> [R2] Abdolmaleki, Abbas, et al. "Maximum a posteriori policy optimisation." arXiv preprint arXiv:1806.06920 (2018).
>
> > Q5. Why $\hat{x}_{0,t}$ can be used in policy improvement, but not policy evaluation.
>
> A5. The intuitive explanation is that the action plays a different role in these two steps. Policy evaluation aims to estimate the Q-value (cumulative reward) of a given action starting from a state. Therefore, a correctly paired action-Q data point is important. However, in policy improvement, we have two targets: 1) behavior cloning by diffusion objective, 2) Q guidance by RL objective (Eq. (7)). The rl objective is used to guide the policy optimization towards actions with high return. Though $\hat{x}_{0,t}$ is not the precise action, it can still provide valuable guidance for policy optimization.
>
> Moreover, we can draw connections to classifier guidance in diffusion models. The Q-network here is a bit like the classifier used in diffusion models. Instead of providing guidance on noisy intermediate steps at inference time, Q-network and action approximation gives guidance at training time.

---

> > ### Comment · Reviewer_TjdE · 2023-08-16
> >
> > I'm disappointed by the author's feedback.
> >
> > - The Diffusion-QL paper repeatedly states its use of K=5 in experiments, e.g., “We found N = 5 performs well on D4RL (Fu et al., 2020) datasets, which is also a small enough value for cost-effective training and deployment. “, and “In the following D4RL tasks, we set a moderate value, N = 5, to balance the performance and computational cost.”. Additionally, the official Twitter-associated Diffusion-QL repository doesn't label itself as "official." You specifically mentioned Twitter-associated repo so you should know they provided another official repo.
> > It's misleading to overstate improvements by selecting hyperparameters for baseline methods. Claims in this paper, especially on efficiency, should be factual.
> > - The response does not make sense to me. In Algorithm 1, there is one section saying, “Sample next actions with DPM-Solver”, the actions samples here need a full reverse process. The statement “our training scheme only passes through the network once an iteration, no matter how big K is” is incorrect.
> > - The authors admit their proposed methods are simple. Actually, there is nothing specific new in the design of diffusion policy. The authors mainly proposed to use the $\hat{a_{0,t}}$ to replace the actual action samples. The likelihood-based extension is not real likelihood estimation.
> > - You know other works have studied EAS, but I didn’t see any references in Section 4.5.
> > - The intuitive explanation does not convince me. You suggest that policy improvement does not need precise action samples. The observation is intriguing but seems counterfactual.
> >
> > In light of these points, my concerns remain. While I recognize the empirical performance, it's vital for claims to be accurate. I'm eager to hear further from the authors.

---

> > > ### Author Response · Authors · 2023-08-17
> > > **Official Comment by Authors [1/2]**
> > >
> > > Dear Reviewer TjdE:
> > >
> > > Thank you very much for your effort at reviewing our paper and valuable comments. We now address your concerns as below.
> > >
> > > > R2-Q1.  You specifically mentioned Twitter-associated repo so you should know they provided another official repo. It's misleading to overstate improvements by selecting hyperparameters for baseline methods.
> > >
> > > We’d like to clarify that we originally intended to paste an url pointing to the code snippet here. However, as external links are strictly prohibited, we break down the url into indicators like organization, repo and file. When we started this project, the only available codebase was under the Twitter organization (the other one was not yet publicly available). We apologize that we did not notice the newly released one has different default parameters.
> > >
> > > In addition, We’d like to clarify that in our experiments, we always emphasize **the configuration of our reimplemented baseline methods** and our improvement over the baseline.
> > >
> > > Moreover, we’d like to emphasize that action approximation brings the following intriguing benefits, which should not be neglected.
> > > Enabling training with large diffusion steps (1000 steps in EDP versus 5 steps in Diffusion-QL).
> > > Large diffusion steps give significant performance boost on kitchen, adroit and antmaze.
> > > Large diffusion steps allow us to use one set of hyperparameters for environments from the same domain.
> > >
> > > We will add further clarifications about the default configurations of Diffusion-QL and the main benefits of using action approximation in our revisions to avoid confusions.
> > >
> > > > R2-Q2. Incorrect statement about “our training scheme only passes through the network once an iteration, no matter how big K is”.
> > >
> > > In terms of this statement, we respectfully disagree. We refer the reviewer to section 4.4, line 217 to 227, where a detailed explanation on the efficiency is provided. A typical actor-critic RL algorithm alternates between policy evaluation and policy improvement, both of which affect the training efficiency. Specifically, *policy evaluation* is used to estimate the state or state-action values for the current policy, *i.e.*, Eqn (6). *Policy improvement* instead focusing on improving the current policy based on the value estimation, as shown in Eqn (7). In section 4.4, we specifically mention that EDP can only reduce the time steps to 15 for policy evaluation (line 218-229). But for policy improvement, EDP does not need backpropagation through the whole sampling chain and only needs to forward and backward the network once per iteration. Please note that our claim is for policy improvement, as explained in line 220-226.
> > >
> > >
> > > > R2-Q3. “Actually, there is nothing specific new in the design of diffusion policy.” “The likelihood-based extension is not real likelihood estimation.”
> > >
> > > We’d like to emphasize that our training strategy enables training diffusion policies with much larger steps, *e.g.*, 1000, which bring clear performance gains on adroit and kitchen environments. Moreover, we are not aiming at estimating the likelihood. Our target is making diffusion policies compatible with likelihood-based RL methods, so that we can approximately maximize the policy likelihood when a diffusion policy is trained with such algorithms. As a result, EDP works nicely with IQL and CRR, which further boosts the performance on antmaze, adroit and kitchen. We argue that the importance of our contributions should not be disregarded.
> > >
> > >
> > > > R2-Q4. “You know other works have studied EAS, but I didn’t see any references in Section 4.5.”
> > >
> > > We thank the reviewer for bringing up this discussion again. We apologize for the confusion caused. Nevertheless, we respectfully disagree with the reviewer, and would like to further clarify on this point. The Energy-based Action Selection (EAS) involves two distinct steps: 1) sampling $K$ actions from $\pi_\theta(a\mid s)$ and 2) selecting the action with the highest $e^{Q(s, a)}$. In fact, the exact form of EAS has been less discussed in the literature and people use it for different purposes. In our case, we use it for reducing the variance of a diffusion policy. Mathematically, such a sampling scheme can be treated as sampling from a non-parametric distribution $p’(a\mid s)\propto e^{Q(s, a)}\pi_\theta(a\mid s)$. Here, we refer to MPO **only to provide intuitive explanations to why EAS improves the final performance**, because $p’(a\mid s)$ can be considered as a one-step improved policy given **current** policy $\pi_\theta(a\mid s)$ and an **estimated** Q values $Q(s, a)$. We **do not suggest that MPO uses EAS for policy improvement**. Specifically, MPO performs policy improvement by minimizing the difference between the current learning policy $\pi_\theta(a\mid s)$ and the non-parametric one-step improved policy $e^{Q(s, a)}\pi(a\mid s)$. Such a scheme is completely different from the EAS. We will further clarify this and improve the presentation in our revisions.

---

> > > ### Author Response · Authors · 2023-08-17
> > > **Official Comment by Authors [2/2]**
> > >
> > >
> > > > R2-Q5. “The intuitive explanation does not convince me. You suggest that policy improvement does not need precise action samples. The observation is intriguing but seems counterfactual.”
> > >
> > > We’d like to emphasize that our intuitive explanations in A5 of the rebuttal are actually two-fold. We first explain why action approximation can not be used for policy evaluation, then explain why it can be used for policy improvement. We did not claim “policy improvement does not need precise action samples”, our point is: compared to policy evaluation, “Though
> > > \hat{x}_{0,t} is not the precise action, it can still provide valuable guidance for policy optimization.” There is some empirical evidence for this from the field of learning from demonstration[R3-R5]. These papers show that when some imperfect demonstration data are added for RL optimization, the performance can be greatly boosted. Moreover, we also draw a connection between our method and classifier guidance, which we believe is able to better understand why action approximation works.
> > >
> > > In the end, we do admit that this intuitive explanation is based on our understanding of RL algorithms, learning from demonstration, offline RL and diffusion models. We do not have theoretical verification for it, but only empirical support that action approximation does not hurt performance much, and it can enable training diffusion models with much larger diffusion steps.
> > >
> > >
> > >
> > > [R3]  Gao, Yang, et al. "Reinforcement learning from imperfect demonstrations." arXiv preprint arXiv:1802.05313 (2018).
> > > [R4] Jing, Mingxuan, et al. "Reinforcement learning from imperfect demonstrations under soft expert guidance." Proceedings of the AAAI conference on artificial intelligence. Vol. 34. No. 04. 2020.
> > > [R5] Kang, Bingyi, Zequn Jie, and Jiashi Feng. "Policy optimization with demonstrations." International conference on machine learning. PMLR, 2018.
> > >
> > > If you have further questions, please do not hesitate to let us know.
> > > Authors of Paper 6867

---

> > > > ### Comment · Reviewer_TjdE · 2023-08-17
> > > >
> > > > Thank you for your continued discussion. While many of my initial concerns have been addressed, I believe the paper could benefit from further clarifications to ensure the accuracy of the claims presented. I've accordingly adjusted my score.
> > > >
> > > > - I noticed no issues with the Github repositories. However, I'd like to highlight that Diffusion-QL explicitly mentioned using K=5 for all their experiments. When your paper reports a 5-day running time of Diffusion-QL based on K=100 and then contrasts it for efficiency, it may come off as an unfair comparison. It would be beneficial to add clarity to this point.
> > > > - For enhanced clarity, it might be useful to specify "in policy improvement" within the relevant sentences.
> > > > - If you're considering EAS as one of the significant contributions of your work, it's essential to provide a comparison with other works and what something new you have contributed.
> > > > - It's worth noting that the use of action approximation and its effectiveness is an interesting observation in your paper.

---

> > > > > ### Author Response · Authors · 2023-08-18
> > > > >
> > > > > Dear Reviewer TjdE,
> > > > >
> > > > > Thank you very much for your quick reply and constructive comments!  We will definitely enhance the clarity of the paper accordingly following your suggestions.
> > > > >
> > > > > Thank you again!
> > > > > Authors of Submission6867

---

### Official Review · Reviewer_tRDF · 2023-07-10

**Soundness:** 4 excellent
**Presentation:** 3 good
**Contribution:** 3 good
**Rating:** 6
**Confidence:** 5

**Summary:**

This paper proposed EDP to address the existing limitation of diffusion policy in offline RL. EDP relies on an action approximation to construct actions from corrupted ones, thus avoiding running the Markov chain for action sampling at training. shows that EDP achieves 25+ speedup over Diffusion-QL at training time on the gym-locomotion tasks in D4RL. Extensive experiments by training EDP with various offline RL algorithms, including TD3, CRR, and IQL justify the superiority of the proposed method.


**Strengths:**

* The proposed method EDP focuses on improving the Diffusion-QL baseline from two aspects including computational cost and limited applicability to different RL algorithms, which is the major contribution of this paper. It provides a new way to apply a complicated generative model in RL.
* Extensive evaluation of datasets of different domains is performed. EDP can outperform several widely-used algorithms by replacing the originally-used Gaussian policy with the refined diffusion policy. And both the training and sampling speed of EDP are way faster than that of Diffusion-QL. Besides, it seems EDP is easier to tune compared with Diffusion-QL as it uses the same hyperparameters for tasks belonging to the same domain and could achieve satisfactory performance.
* The unreasonableness of evaluation protocol OMS is pointed out and RAT is adopted to justify the performance of the proposed method and baselines.


**Weaknesses:**

* The compared baselines  in Table 2 and Figure 1 are a little bit weak, which cannot demonstrate the superiority of EDP.  As the authors claimed that the proposed method outperforms pre-sota methods, more powerful model-free baselines should be included such as EDAC[1], RORL[2] on Locomotion tasks, X-QL[3], InAC[4], BPPO[5] on other domains. Otherwise, I believed the superiority of the proposed method is overclaimed to some extent.

  [1] An, Gaon, et al. "Uncertainty-based offline reinforcement learning with diversified q-ensemble." *Advances in neural information processing systems* 34 (2021): 7436-7447.

  [2] Yang, Rui, et al. "Rorl: Robust offline reinforcement learning via conservative smoothing." *Advances in Neural Information Processing Systems* 35 (2022): 23851-23866.

  [3] Garg, Divyansh, et al. "Extreme Q-Learning: MaxEnt RL without Entropy." *The Eleventh International Conference on Learning Representations*. 2022.

  [4] Xiao, Chenjun, et al. "The In-Sample Softmax for Offline Reinforcement Learning." *The Eleventh International Conference on Learning Representations*. 2022.

  [5] Zhuang, Zifeng, et al. "Behavior Proximal Policy Optimization." *The Eleventh International Conference on Learning Representations*. 2022.


**Questions:**

* It seems there exist mismatch of results between pre-sota  methods from Figure 1 and Table 2?
* Line 227 points out that Tab 1 reflects the benefits of increasing K. However, it seems Tab 1 doesn't include this information?
* The GPU memory needed by Diffusion-QL is quite large. I'm curious whether EDP has the same problem?

---

> ### Author Rebuttal · Authors · 2023-08-09
>
> > Q1. The baselines are a little bit weak. The superiority might be overclaimed.
>
> A1. Thank you for reminding us of these important literatures.  We’d like to clarify that our paper focuses on policy representation in offline RL, which is orthogonal to the algorithmic developments by these related works. It means that our EDP can be further integrated into their methods for better policy modeling. Moreover, we have tried our best to make a fair comparison with these methods during rebuttal period. Unfortunately, we failed to do so due to the mismatch of evaluation protocols, network architectures and hyper-parameters. Taking BPPO as an example, in their Figure 3 (c), the running average score for  hopper-medium-expert-v2 is around 90-100, while in Table 1, the OMS score is reported as 112.8.
>
> In light of the aforementioned challenges and your insightful suggestion, we are happy to adjust our claim to make it more precise. We will restrict the sota claim to the policy representation scope.
>
> > Q2. Mismatch of pre-sota methods from Figure 1 and Table 2.
>
> A2. Thank you so much for pointing out this typo. We will update Figure 1 to make the pre-sota of Adroit be 65.9 instead of the current 56.9.
>
> > Q3. The benefits of increasing K
>
> A3. Sorry for the confusion.  We thank you for the valuable question and will definitely improve this in our revision. In table 1, Diffusion-QL are directly copied from the original paper, which is obtained by setting K to 5. DQL(JAX) is our reimplementation of Diffusion-QL but with K equals to 100. EDP is our final version whose training speed is constant to the number of K (thanks to action approximation). Therefore, EDP is using 1000 for K. On all 4 domains, EDP surpasses Diffusion-QL, which means larger K brings better performance.
>
> > Q4. The GPU memory needed by Diffusion-QL is quite large, how about EDP?
>
> A4. EDP does not suffer from this problem. The reason Diffusion-QL needs large GPU memory is that it needs to forward and backward the policy network K times to sample an action at each training iteration. However, EDP avoids this problem by introducing the action approximation technique, which only forwards and backwards the network once, thus greatly saving GPU memory usage and training time.

---

> ### Comment · Area_Chair_fc4X · 2023-08-16
> **Reviewering Input Needed**
>
> Hello Reviewers,
>
> The authors have made efforts to address your comments on their work via the rebuttal. Part of the NeurIPS review process is participating meaningfully in the rebuttal phase to help ensure quality. Please read and respond to the author's comments today, latest tomorrow, to give everyone time to respond and reach proper conclusions.
>
> Thank you all again for your assistance in making NeurIPS a great conference for our community.

---

> > ### Comment · Area_Chair_fc4X · 2023-08-18
> > **Reviewer Response Needed**
> >
> > Hello Reviewer,
> >
> > The authors have made efforts to address your comments on their work via the rebuttal. Part of the NeurIPS review process is participating meaningfully in the rebuttal phase to help ensure quality. Please read and respond to the author's comments today, latest tomorrow, to give everyone time to respond and reach proper conclusions.
> >
> > Thank you for your assistance in making NeurIPS a great conference for our community.
> > -- Your AC

---

### Author Rebuttal · Authors · 2023-08-10

We sincerely thank all the reviewers for recognizing the novelty and contributions of our work, as well as for providing valuable questions for discussion and constructive suggestions.

As there are limited shared questions from the reviewers, we address them individually in the corresponding responses. Specifically, as requested by Reviewer 44Ux and Reviewer SXfY, we added additional experiments that we'd like to attract your attention, including

- ablating the policy approximation by comparing Eqn. 12 and Eqn. 13 in likelihood-based policy optimization (requested by Reviewer 44Ux).
- ablating the effect of action approximation (requested by Reviewer SXfY)
- ablating the effect of the diffusion timesteps K (requested by Reviewer SXfY)

We have directly appended the updated tables to our responses for the respective reviewers. We value your feedback and please kindly let us know if you have any further questions or concerns.

Sincerely,

Authors

---

### Decision · Program_Chairs · 2023-09-21

**Decision:**

Accept (poster)

**Comment:**

This work showcases a new implementation of a diffusion policy that is algorithm-independent. The reviewers agree this paper shows a contribution across algorithms. The use of DPM-Solver is highlighted as a major contribution in speeding up the sampling process compared to DDPM sampling in Diffusion-QL. There are other algorithms for improving the sampling speed for DPM solvers. The proposed method works well for RL problems. I also tend to agree the breadth of experiments shows a general improvement.